# Quantum Vacuum Effects in Braneworlds on AdS Bulk

**Aram A. Saharian** 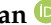

Department of Physics, Yerevan State University, 1 Alex Manoogian Street, Yerevan 0025, Armenia; saharian@ysu.am

**Abstract:** We review the results of investigations for brane-induced effects on the local properties of quantum vacuum in background of AdS spacetime. Two geometries are considered: a brane parallel to the AdS boundary and a brane intersecting the AdS boundary. For both cases, the contribution in the vacuum expectation value (VEV) of the energy–momentum tensor is separated explicitly and its behavior in various asymptotic regions of the parameters is studied. It is shown that the influence of the gravitational field on the local properties of the quantum vacuum is essential at distance from the brane larger than the AdS curvature radius. In the geometry with a brane parallel to the AdS boundary, the VEV of the energy–momentum tensor is considered for scalar field with the Robin boundary condition, for Dirac field with the bag boundary condition and for the electromagnetic field. In the latter case, two types of boundary conditions are discussed. The first one is a generalization of the perfect conductor boundary condition and the second one corresponds to the confining boundary condition used in QCD for gluons. For the geometry of a brane intersecting the AdS boundary, the case of a scalar field is considered. The corresponding energy–momentum tensor, apart from the diagonal components, has nonzero off-diagonal component. As a consequence of the latter, in addition to the normal component, the Casimir force acquires a component parallel to the brane.

**Keywords:** anti-de Sitter spacetime; quantum vacuum; braneworld; Casimir effect

## 1. Introduction

Quantum field theory in anti-de Sitter (AdS) spacetime is an active area of research. This activity is motivated by several reasons. First of all, the corresponding geometry is maximally symmetric and a sufficiently large number of problems are exactly solvable on its background. This helps to give an idea of the influence of the classical gravitational field on quantum phenomena in more complicated geometries. Qualitatively new features in the dynamics of quantum fields on the AdS background are related to the lack of global hyperbolicity and to the existence of both regular and irregular modes. In particular, boundary conditions on propagating fields need to be imposed at timelike infinity to prevent loss of unitarity. The different boundary conditions define different theories in the bulk. Another new feature, which has no analog in Minkowskian field theories, is related to the possibility of regularization for infrared divergences in interacting field theories without reducing the symmetries [1]. This is closely related to the natural length scale of the AdS spacetime. The high interest to the AdS geometry is also related to its natural appearance as a ground state in supergravity and as the near horizon geometry for extremal black holes, black strings and domain walls.

The AdS spacetime plays a fundamental role in two exciting developments of contemporary theoretical physics. The first one, the AdS/CFT correspondence (for reviews, see [2–4]), establishes duality between string theories or supergravity in the AdS bulk and a conformal field theory localized on the AdS boundary.

This duality provides an interesting possibility for the investigation of nonperturbative effects in both sides of the correspondence by using the weak coupling regime of the dual theory. The recent developments include applications in various condensed matter systems such as holographic superconductors and topological insulators [5,6]. The second development is related to various types of braneworld models with large extra dimensions [7]. In the corresponding setup, the standard model fields are localized on a brane embedded in a higher dimensional AdS spacetime. Braneworlds naturally appear in string/M-theory context and have been initially proposed for a geometrical resolution of the hierarchy problem between the electroweak and Planck energy scales. They provide an alternative framework to address the problems in particle physics and cosmology from different perspectives.

An inherent feature of field theories in AdS/CFT correspondence and in braneworld models is the need to impose boundary conditions on fields propagating in the AdS bulk. They include the conditions on the AdS boundary and the conditions on the branes in braneworld scenario. In particular, in braneworld models of the Randall–Sundrum type the boundary conditions on the branes are dictated by the $Z_2$-symmetry. In quantum field theory, the boundary conditions modify the spectrum of the zero-point fluctuations and, as a consequence, the vacuum expectation values (VEVs) are shifted by an amount that depends on the bulk and boundary geometries and also on the boundary conditions. This is the well-known Casimir effect [8–11]. In braneworld models the Casimir forces acting on the branes may provide a mechanism for stabilization of the brane location (for mechanisms of moduli stabilization in warped geometries, see, e.g., [12–18] and references therein). In particular, this stabilization is required to prevent the variations of physical constants on the branes. In addition, the quantum effects of bulk fields generate a cosmological constant on the brane. Motivated by these points, the Casimir effect in braneworld models on the AdS bulk, with branes parallel to the AdS boundary, has been investigated for scalar [19–38], fermionic [39–43] and vector fields [44–49]. The models with de Sitter branes have been discussed in [50–58]. The brane-induced quantum vacuum effects in AdS spacetime with additional compact subspaces were considered in [59–65].

The main part of the papers on the Casimir effect in the AdS bulk consider global quantities, such as the Casimir energy and the forces acting on the branes. The local quantities carry more detailed information about the properties of the quantum vacuum. In particular, the expectation value of the energy–momentum tensor is of special interest. It appears as the source in the semiclassical Einstein equations and therefore plays an important role in modeling self-consistent dynamics involving the gravitational field. The VEV of the energy–momentum tensor for a conformally coupled scalar field in conformally-flat geometries has been investigated in [28]. Massive scalar fields with general curvature coupling in the geometry of two branes on AdS bulk were considered in [66,67]. The Casimir densities for a $Z_2$-symmetric thick brane for the general case of static plane symmetric interior structure have been discussed in [68]. The VEVs of the energy–momentum tensor for Dirac spinor field and for the electromagnetic field are investigated in [42,43,46–49]. The geometry of a brane intersecting the AdS boundary is considered in [69]. For a scalar field with general curvature coupling, the background AdS geometry with an additional compact subspace is discussed in [61,62].

In the present paper, we review the results for the VEV of the energy–momentum tensor in the geometry of a single brane on background of $(D + 1)$-dimensional AdS spacetime. The cases of scalar, Dirac and electromagnetic fields are considered. The organization of the paper is as follows. In Section 2, for a planar brane parallel to the AdS boundary, we consider complete sets of orthonormalized mode functions for both the regions between the AdS boundary and the brane and between the brane and the AdS horizon. The VEVs of the energy–momentum tensors for scalar, Dirac and electromagnetic fields are investigated in Section 3. The behavior in the asymptotic regions of the parameters is discussed in detail. Section 4 considers the VEV of the surface energy–momentum tensor for a scalar field on a brane parallel to the AdS boundary. Section 5 is devoted to the study of the vacuum energy–momentum tensor

for a scalar field in the geometry with a brane perpendicular to the AdS boundary. The main results are summarized in Section 6.

## 2. Mode Functions in the Geometry with a Brane Parallel to the AdS Boundary

In Poincaré coordinates $(x^0 = t, x^1, \ldots, x^{D-1}, y)$, with $-\infty < x^i, y < +\infty$, $i = 0, 1, \ldots, D-1$, the line element for the $(D+1)$-dimensional AdS spacetime is presented as

$$ds^2 = e^{-2y/a}\left[\left(dx^0\right)^2 - \left(dx^1\right)^2 - \cdots - \left(dx^{D-1}\right)^2\right] - dy^2, \tag{1}$$

where $a$ is the curvature radius of the background geometry sourced by a negative cosmological constant $\Lambda = -D(D-1)/(2a^2)$. For the curvature scalar and the Ricci tensor, one has $R = -D(D+1)/a^2$ and $R_{\mu\rho} = -Dg_{\mu\rho}/a^2$, with the metric tensor $g_{\mu\rho}$ defined from (1). Introducing a new coordinate $z$, $0 \leqslant z < \infty$, in accordance with $z = ae^{y/a}$, the line element is written in conformally flat form

$$ds^2 = g_{\mu\rho}dx^\mu dx^\rho = (a/z)^2 \eta_{\mu\rho}dx^\mu dx^\rho, \tag{2}$$

where $x^D = z$ and $\eta_{\mu\rho} = \mathrm{diag}(1, -1, \ldots, -1)$ is the metric tensor for $(D+1)$-dimensional Minkowski spacetime. The hypersurfaces $z = 0$ and $z = \infty$ correspond to the AdS boundary and horizon, respectively. In what follows, we work in the coordinate system defined by (2).

We are interested in the effects on the local properties of the quantum vacuum induced by a codimension one brane. First, we consider the case where the brane is parallel to the AdS boundary and is located at $z = z_0$. It divides the space into two subspaces: the region between the AdS boundary and the brane, $0 \leq z \leq z_0$ (L-region), and the region between the brane and AdS horizon, $z_0 \leq z < \infty$ (R-region). The brane has a nonzero extrinsic curvature tensor and, as a consequence, the properties of the vacuum state in the L- and R-regions are different. The evaluation of the VEVs for local physical observables in those regions requires different procedures and we will discuss them separately. The VEVs are presented in the form of mode-sums over complete set of mode functions for quantum fields and we start by considering the modes for scalar, Dirac and electromagnetic fields.

### 2.1. Scalar Field

First, we consider a scalar field $\varphi(x)$ with the mass $m$. Assuming a general curvature coupling with the parameter $\xi$, the field equation reads

$$\left(g^{\mu\rho}\nabla_\mu\nabla_\rho + m^2 + \xi R\right)\varphi(x) = 0, \tag{3}$$

where $\nabla_\mu$ is the covariant derivative operator. The most important special cases correspond to minimally ($\xi = 0$) and conformally ($\xi = \xi_D = (D-1)/(4D)$) coupled fields. Let $\varphi_\sigma^{(\pm)}(x)$ be a complete set of positive and negative energy mode functions specified by the set of quantum numbers $\sigma$. In accordance with the problem symmetry, they can be presented in the form

$$\varphi_\sigma(x) = e^{i\mathbf{kx}\mp i\omega t}f(z), \tag{4}$$

where $\mathbf{x} = (x^1, x^2, \ldots, x^{D-1})$, $\mathbf{k} = (k_1, k_2, \ldots, k_{D-1})$, and $\mathbf{kx} = \sum_{l=1}^{D-1} k_i x^i$. Plugging into the field equation, we get an ordinary differential equation for the function $f(z)$:

$$z^{D+1}\partial_z\left[z^{1-D}\partial_z f(z)\right] + \left[\lambda^2 z^2 + \left(\xi D(D+1) - a^2 m^2\right)\right]f(z) = 0, \tag{5}$$

where the energy $\omega$ is expressed in terms of $\lambda$ and $k^2 = \sum_{i=1}^{D-1} k_i^2$ as $\omega = \sqrt{\lambda^2 + k^2}$. The general solution of (5) is expressed in terms of the Bessel and Neumann functions $J_\nu(\lambda z)$ and $Y_\nu(\lambda z)$:

$$f(z) = z^{D/2} \left[ c_1 J_\nu(\lambda z) + c_2 Y_\nu(\lambda z) \right], \tag{6}$$

where $c_1$ and $c_2$ are constants and

$$\nu = \sqrt{D^2/4 - D(D+1)\xi + m^2 a^2}. \tag{7}$$

To have a stable vacuum we assume the values of the parameters in (7) for which $\nu \geqslant 0$ (see [70,71]). For a conformally coupled massless field, one has $\nu = 1/2$ and the mode functions (4) with (6) are conformally related to the modes in the Minkowski bulk. The scalar modes are normalized by the condition

$$\int d^D x \, g^{00} \sqrt{|g|} \varphi_\sigma^{(s)}(x) \varphi_{\sigma'}^{(s')*}(x) = \frac{\delta_{ss'}}{2\omega} \delta_{\lambda\lambda'} \delta(\mathbf{k} - \mathbf{k}'), \tag{8}$$

where $g$ is the determinant of the metric tensor. Here, $\delta_{\lambda\lambda'}$ is the Kronecker delta in the problems with discrete eigenvalues of the quantum number $\lambda$ and $\delta_{\lambda\lambda'} = \delta(\lambda - \lambda')$ in the problems with continuous spectrum for $\lambda$.

We are interested in the effects of a codimension one brane, localized at $z = z_0$, on the local properties of the scalar vacuum. The Robin boundary condition is imposed for the field operator on the brane:

$$(\beta n^\mu \nabla_\mu + 1)\varphi(x) = 0, \ z = z_0, \tag{9}$$

where $n^\mu$ is the inward pointing normal to the brane and $\beta$ is a constant. The latter encodes the properties of the brane. In the special cases $\beta = 0$ and $\beta = \infty$, the condition (9) is reduced to the Dirichlet and Neumann boundary conditions, respectively. For the normal in (9) one has $n^\mu = \delta_{(J)} \delta_D^\mu z/a$, where J = L, $\delta_{(L)} = -1$ in the L-region and J = R, $\delta_{(R)} = 1$ in the R-region. In general, the values of the constant $\beta$ could be different for those regions.

First, let us consider the modes in the R-region. From the boundary condition (9), it follows that $c_2/c_1 = -\bar{J}_\nu(\lambda z_0)/\bar{Y}_\nu(\lambda z_0)$ for the coefficients in (6). Here and below, for a given function $F(x)$, we use the notation with the bar defined in accordance with

$$\bar{F}(x) = B_0 x F'(x) + A_0 F(x), \tag{10}$$

with the coefficients

$$A_0 = 1 + \delta_{(J)} \frac{D\beta}{2a}, \quad B_0 = \delta_{(J)} \frac{\beta}{a}. \tag{11}$$

The mode functions in the R-region obeying the boundary condition (9) are presented as

$$\varphi_{(R)\sigma}^{(\pm)}(x) = C_{(R)\sigma} z^{D/2} g_\nu(\lambda z_0, \lambda z) e^{i\mathbf{k}\mathbf{x} \mp i\omega t}, \tag{12}$$

with the function

$$g_\nu(u, v) = J_\nu(v) \bar{Y}_\nu(u) - \bar{J}_\nu(u) Y_\nu(v). \tag{13}$$

The spectrum for $\lambda$ is continuous, and, from the normalization condition (8) with $\delta_{\lambda\lambda'} = \delta(\lambda - \lambda')$, we get

$$|C_{(R)\sigma}|^2 = \frac{(2\pi a)^{1-D} \lambda}{2\omega \left[ \bar{J}_\nu^2(\lambda z_0) + \bar{Y}_\nu^2(\lambda z_0) \right]}. \tag{14}$$

The modes are specified by the set $\sigma = (\mathbf{k}, \lambda)$ with $-\infty < k_i < +\infty$, $i = 1, \ldots, D - 1$, and $0 \leq \lambda < \infty$. The analog of the mode functions (12) in the region between two parallel branes on the AdS bulk has been considered in [67].

Note that we could also have modes with purely imaginary $\lambda$, $\lambda = i|\lambda|$. For those modes, $f(z) = c_1 z^{D/2} K_\nu(|\lambda|z)$, where $K_\nu(x)$ is the MacDonald function (the modes with the modified Bessel function $I_\nu(|\lambda|z)$ are not normalizable). From the boundary condition (9), we get the equation for the allowed values of $|\lambda|$: $\bar{K}_\nu(|\lambda|z_0) = 0$. The energy corresponding to these modes is given by $\omega = \sqrt{k^2 - |\lambda|^2}$ and it becomes imaginary for $k < |\lambda|$. This leads to the instability of the vacuum state. To exclude the unstable modes, we restrict the allowed values for the Robin coefficient in the region where the equation $\bar{K}_\nu(|\lambda|z_0) = 0$ has no roots. It can be seen that, for non-Dirichlet boundary conditions ($\beta \neq 0$), the corresponding condition is expressed as $a/\beta < \nu - D/2$ (for more detailed discussion in models with compact dimensions, see [72]). For $a/\beta > \nu - D/2$ there is a single root. For the special value $\beta/a = 1/(\nu - D/2)$, there exists a mode with $\lambda = 0$ and with the mode functions $\varphi_{(R)\sigma}^{(\pm)}(x) = C_{(R)\sigma} z^{D/2-\nu} e^{i\mathbf{kx} \mp ikt}$. For a minimally coupled massless scalar field, one has $\nu = D/2$ and this special value corresponds to the Neumann boundary condition. The corresponding mode functions do not depend on $z$.

In the L-region, the integration over $z$ in (8) goes over the interval $z \in [0, z_0]$. For the solutions (4) with (6) and $c_2 \neq 0$, the $z$-integral diverges at the lower limit $z = 0$ in the range $\nu \geq 1$. Hence, in that range, from the normalizability condition for the mode functions it follows that we should take $c_2 = 0$. In the range $0 \leq \nu < 1$, the solution (6) with $c_2 \neq 0$ is normalizable and in order to uniquely define the mode functions an additional boundary condition at the AdS boundary is required [70,73]. The general class of allowed boundary conditions has been discussed in [74,75]. In particular, they include the Dirichlet and Neumann boundary conditions, the most frequently used in the literature. Here, for the values of the parameters corresponding to the range $0 \leq \nu < 1$, we choose the Dirichlet condition, which gives $c_2 = 0$. With this choice, the mode functions in the L-region are specified as

$$\varphi_{(L)\sigma}^{(\pm)}(x) = C_{(L)\sigma} z^{D/2} J_\nu(\lambda z) e^{i\mathbf{kx} \mp i\omega t}. \tag{15}$$

From the boundary condition (9) on the brane we get the equation for the eigenvalues of the quantum number $\lambda$:

$$\bar{J}_\nu(\lambda z_0) = 0. \tag{16}$$

If we denote by $x = \lambda_{\nu,n}$ the positive zeros of the functions $\bar{J}_\nu(x)$, numerated by $n = 1, 2, \ldots$, then the eigenvalues are given by $\lambda = \lambda_{\nu,n}/z_0$. Note that the roots $\lambda_{\nu,n}$ do not depend on the location of the brane. From the normalization condition (8), with $\delta_{\lambda\lambda'} = \delta_{nn'}$ and with the $z$-integration over $[0, z_0]$, one finds

$$|C_\sigma|^2 = \frac{(2\pi a)^{1-D} \lambda_{\nu,n} T_\nu(\lambda_{\nu,n})}{z_0 \sqrt{k^2 z_0^2 + \lambda_{\nu,n}^2}}, \tag{17}$$

with the function $T_\nu(x) = x[x^2 J_\nu'^2(x) + (x^2 - \nu^2) J_\nu^2(x)]^{-1}$.

Similar to the R-region, the stability condition for the vacuum state in the L-region imposes restrictions to the allowed values of the Robin coefficient $\beta$. That condition excludes the presence of purely imaginary roots $\lambda = i|\lambda|$ for the Equation (16). It can be shown that there are no such modes for $a/\beta < D/2 + \nu$ and there is a single mode for $a/\beta > D/2 + \nu$. As seen, the stability condition in the L-region is less restrictive than that for the R-region. In the special case $\beta/a = 1/(D/2 + \nu)$ one has a mode with $\lambda = 0$ with the mode functions $\varphi_{(L)\sigma}^{(\pm)}(x) = C_{(L)\sigma} z^{D/2+\nu} e^{i\mathbf{kx} \mp ikt}$.

## 2.2. Dirac Field

Now, we turn to a massive Dirac field $\psi(x)$. The dynamics of the field is governed by the Dirac equation

$$\left[ i\gamma^\mu \left( \partial_\mu + \Gamma_\mu \right) - m \right] \psi(x) = 0, \tag{18}$$

where $\Gamma_\mu$ is the spin connection. The curved spacetime Dirac matrices $\gamma^\mu$ are expressed in terms of the flat spacetime matrices $\gamma^{(b)}$ by the relation $\gamma^\mu = e^\mu_{(b)} \gamma^{(b)}$, with $e^\mu_{(b)}$ being the tetrad fields. In the coordinate system corresponding to (2), the latter can be taken as $e^\mu_{(b)} = (z/a)\delta^\mu_b$. For the components of the spin connection, this gives $\Gamma_D = 0$ and $\Gamma_i = \eta_{il}\gamma^{(D)}\gamma^{(l)}/(2z)$ for $i = 0, \ldots, D - 1$. In an irreducible representation of the Clifford algebra, the matrices $\gamma^{(b)}$ are $N \times N$ matrices, where $N = 2^{[(D+1)/2]}$ and the square brackets in the exponent mean the integer part. Up to a similarity transformation, the irreducible representation is unique in odd numbers of spatial dimension $D$. For even values of $D$, one has two inequivalent irreducible representations. We use the flat spacetime gamma matrices in the representation

$$\gamma^{(0)} = \begin{pmatrix} 0 & \chi_0 \\ \chi_0^\dagger & 0 \end{pmatrix}, \ \gamma^{(l)} = \begin{pmatrix} 0 & \chi_l \\ -\chi_l^\dagger & 0 \end{pmatrix}, \ l = 1, 2, \ldots, D - 1, \tag{19}$$

and $\gamma^{(D)} = si \, \text{diag}(1, -1)$, where $s = \pm 1$. In odd spatial dimensions, the two values of the parameter $s$ correspond to two inequivalent representations. The commutation relations for the $N/2 \times N/2$ matrices $\chi_l$ and for their hermitian conjugate matrices $\chi_l^\dagger$ are obtained from those for the Dirac matrices $\gamma^{(l)}$. They are reduced to the relations $\chi_0\chi_l^\dagger = \chi_l\chi_0^\dagger$, $\chi_0^\dagger\chi_l = \chi_l^\dagger\chi_0$, $\chi_0^\dagger\chi_0 = 1$ and $\chi_l\chi_i^\dagger + \chi_i\chi_l^\dagger = 2\delta_{li}$, $\chi_l^\dagger\chi_i + \chi_i^\dagger\chi_l = 2\delta_{li}$, for $l, i = 1, 2, \ldots, D - 1$. The representation (19) for the construction of the gamma matrices in AdS spacetime is considered in [76]. Another representation is taken in [43].

Assuming the dependence on the coordinates $(t, \mathbf{x})$ in the form $e^{i\mathbf{k}\mathbf{x}\mp i\omega t}$ and decomposing the spinor $\psi(x)$ into the upper and lower components, in the representation (19), the corresponding equations are separated. The dependence of those components on the $z$-coordinate is expressed in terms of the function $c_1 J_{ma\pm s/2}(\lambda z) + c_2 Y_{ma\pm s/2}(\lambda z)$, where the upper and lower signs correspond to the upper and lower components. The coefficients are determined by the normalization condition and by the boundary condition on the brane at $z = z_0$. The positive and negative energy fermionic modes $\psi_\sigma^{(\pm)}(x)$, specified by the set of quantum numbers $\sigma$, are normalized by the condition

$$\int d^D x \, (a/z)^D \psi_\sigma^{(\pm)\dagger} \psi_{\sigma'}^{(\pm)} = \delta_{\sigma\sigma'}. \tag{20}$$

As in the case of a scalar field, here $\delta_{\sigma\sigma'}$ is understood as the Dirac delta function for the continuous components of $\sigma$ and the Kronecker delta for discrete ones. On the brane at $z = z_0$, we impose the bag boundary condition

$$(1 + i\gamma^\mu n_\mu)\psi(x) = 0, \ z = z_0, \tag{21}$$

where $n_\mu = -\delta_{(J)}\delta^D_\mu a/z$, J=R,L, with $\delta_{(R)} = -\delta_{(L)} = 1$ for the L- and R-regions.

In the R-region, $z_0 \leq z < \infty$, from the boundary condition (21), for the ratio of the coefficients in the linear combination of the Bessel and Neumann functions, one finds $c_2/c_1 = -J_{ma+1/2}(\lambda z_0)/Y_{ma+1/2}(\lambda z_0)$. The positive and negative energy mode functions, obeying the boundary condition (21), are expressed as

$$
\psi^{(+)}_{(R)\sigma}(x) = C^{(+)}_{(R)\sigma} z^{\frac{D+1}{2}} e^{i\mathbf{kx}-i\omega t}
\begin{pmatrix}
\frac{\mathbf{k\chi}\chi_0^\dagger + i\lambda - \omega}{\omega} g_{ma+1/2, ma+s/2}(\lambda z_0, \lambda z) w^{(\gamma)} \\
i\chi_0^\dagger \frac{\mathbf{k\chi}\chi_0^\dagger + i\lambda + \omega}{\omega} g_{ma+1/2, ma-s/2}(\lambda z_0, \lambda z) w^{(\gamma)}
\end{pmatrix},
$$

$$
\psi^{(-)}_{(R)\sigma}(x) = C^{(-)}_{(R)\sigma} z^{\frac{D+1}{2}} e^{i\mathbf{kx}+i\omega t}
\begin{pmatrix}
i\chi_0 \frac{\mathbf{k\chi}^\dagger \chi_0 - i\lambda + \omega}{\omega} g_{ma+1/2, ma+s/2}(\lambda z_0, \lambda z) w^{(\gamma)} \\
\frac{\mathbf{k\chi}^\dagger \chi_0 - i\lambda - \omega}{\omega} g_{ma+1/2, ma-s/2}(\lambda z_0, \lambda z) w^{(\gamma)}
\end{pmatrix},
\tag{22}
$$

where $\mathbf{k\chi} = \sum_{l=1}^{D-1} k_l \chi_l$ and

$$
g_{\mu,\rho}(x, u) = J_\mu(x) Y_\rho(u) - J_\rho(u) Y_\mu(x).
\tag{23}
$$

In (22), the one-column matrices $w^{(\gamma)}$, $\gamma = 1, \ldots, N/2$ are introduced with the elements $w_l^{(\gamma)} = \delta_{l\gamma}$ and having $N/2$ rows. The normalization constant is determined from the condition (20):

$$
\left| C^{(\pm)}_{(R)\sigma} \right|^2 = \lambda \frac{\left[ J^2_{ma+1/2}(\lambda z_0) + Y^2_{ma+1/2}(\lambda z_0) \right]^{-1}}{4(2\pi)^{D-1} a^D}.
\tag{24}
$$

The set of quantum numbers $\sigma$ is specified as $\sigma = (\mathbf{k}, \lambda, \gamma)$.

In the L-region and for $ma \geq 1/2$, from the normalizability condition, we get $c_2 = 0$. In the range of the mass corresponding to $ma < 1/2$, the modes with $c_2 \neq 0$ are normalizable as well, and an additional boundary condition is required to uniquely define the mode functions. Here, we consider a special case that corresponds to the choice $c_2 = 0$ for all values of the mass. The fermionic modes are given by the expressions

$$
\psi^{(+)}_{(L)\sigma}(x) = C^{(+)}_{(L)\sigma} z^{\frac{D+1}{2}} e^{i\mathbf{kx}-i\omega t}
\begin{pmatrix}
\frac{\mathbf{k\chi}\chi_0^\dagger + i\lambda - \omega}{\omega} J_{ma+s/2}(\lambda z) w^{(\gamma)} \\
i\chi_0^\dagger \frac{\mathbf{k\chi}\chi_0^\dagger + i\lambda + \omega}{\omega} J_{ma-s/2}(\lambda z) w^{(\gamma)}
\end{pmatrix},
$$

$$
\psi^{(-)}_{(L)\sigma}(x) = C^{(-)}_{(L)\sigma} z^{\frac{D+1}{2}} e^{i\mathbf{kx}+i\omega t}
\begin{pmatrix}
i\chi_0 \frac{\mathbf{k\chi}^\dagger \chi_0 - i\lambda + \omega}{\omega} J_{ma+s/2}(\lambda z) w^{(\gamma)} \\
\frac{\mathbf{k\chi}^\dagger \chi_0 - i\lambda - \omega}{\omega} J_{ma-s/2}(\lambda z) w^{(\gamma)}
\end{pmatrix}.
\tag{25}
$$

The allowed values of $\lambda$ are determined from the boundary condition (21) on the brane. They are roots of the equation

$$
J_{ma-1/2}(\lambda z_0) = 0,
\tag{26}
$$

and are expressed as $\lambda = \lambda_{ma-1/2,n}/z_0$. The normalization coefficient is obtained from (20) and is given by

$$
|C^{(\pm)}_{(L)\sigma}|^2 = \frac{J^{-2}_{ma+1/2}(\lambda_{ma-1/2,n})}{2(2\pi)^{D-1} a^D z_0^2}.
\tag{27}
$$

Note that the eigenvalues for $\lambda$ are the same for both the representations of the Clifford algebra.

### 2.3. Electromagnetic Field

For the electromagnetic field with the vector potential $A_\mu(x)$, $\mu = 0, 1, \ldots, D$, the field equation reads

$$
\nabla_\rho F^{\mu\rho} = 0,
\tag{28}
$$

where $F_{\mu\rho} = \partial_\mu A_\rho - \partial_\rho A_\mu$ is the field strength tensor. To find a complete set of mode functions $A_{\sigma\mu}(x)$ for the vector potential, we impose the Lorenz condition $\nabla_\mu A^\mu = 0$ and an additional gauge condition $A^D = 0$. For the positive energy modes, presenting the dependence on the coordinates $(t, x^1, \ldots, x^{D-1})$ in the form $e^{i\mathbf{kx} - i\omega t}$, from the field equation, we can see that

$$A_{\sigma\mu}(x) = \epsilon_{(\gamma)\mu} z^{D/2-1} \left[ c_1 J_{D/2-1}(\lambda z) + c_2 Y_{D/2-1}(\lambda z) \right] e^{i\mathbf{kx} - i\omega t}. \tag{29}$$

Here, $\gamma = 1, \ldots, D-1$ correspond to different polarizations specified by the polarization vector $\epsilon_{(\gamma)\mu}$. For the latter, one has the normalization condition $\eta^{\mu\rho} \epsilon_{(\gamma)\mu} \epsilon_{(\gamma')\rho} = -\delta_{\gamma\gamma'}$ and the constraints $\epsilon_{(\gamma)D} = 0$ and $\eta^{\mu\rho} k_\mu \epsilon_{(\gamma)\rho} = 0$. The latter two relations follow from the gauge conditions. The modes (29) are normalized by the condition

$$\int d^D x \sqrt{|g|} [A^*_{\sigma'\mu} \nabla^0 A^\mu_\sigma - (\nabla^0 A^*_{\sigma'\mu}) A^\mu_\sigma] = 4i\pi\delta_{\sigma\sigma'}, \tag{30}$$

where the set of quantum numbers is given by $\sigma = (\mathbf{k}, \lambda, \gamma)$.

The coefficients $c_1$ and $c_2$ in (29) are determined from (30) and from the boundary condition on the brane. We consider two types of gauge invariant constraints. The first condition is the analog of the perfect conductor boundary condition in $D = 3$ Maxwell electrodynamics and is given by

$$n^{\mu_1} {}^*F_{\mu_1 \cdots \mu_{D-1}} = 0, \; z = z_0, \tag{31}$$

with ${}^*F_{\mu_1 \cdots \mu_{D-1}}$ being the dual of the field tensor and $n^\mu$ is the normal vector to the boundary. The second boundary conditions is expressed as

$$n^\mu F_{\mu\rho} = 0, \; z = z_0. \tag{32}$$

This type of condition has been used in quantum chromodynamics to confine the gluons. In the gauge under consideration and for the mode functions (29), the boundary condition (31) yields $A_{\sigma D}|_{z=z_0} = 0$, whereas the condition (32) gives $\partial_D A_{\sigma l}|_{z=z_0} = 0$.

In the R-region, from the boundary condition on the brane, we get $c_2/c_1 = -J_\nu(\lambda z_0)/Y_\nu(\lambda z_0)$, where

$$\nu = \begin{cases} D/2 - 1, & \text{for (31)}, \\ D/2 - 2, & \text{for (32)}. \end{cases} \tag{33}$$

The corresponding mode functions for the vector potential are presented as

$$A_{\sigma\mu} = C_{(\mathrm{R})\sigma} \epsilon_{(\gamma)\mu} z^{D/2-1} g_{\nu, D/2-1}(\lambda z_0, \lambda z) e^{i\mathbf{kx} - i\omega t}, \tag{34}$$

where the function $g_{\mu,\rho}(x, u)$ is given by (23). The normalization constant is found from (30):

$$C^2_{(\mathrm{R})\sigma} = \lambda \frac{\left[ J^2_\nu(\lambda z_0) + Y^2_\nu(\lambda z_0) \right]^{-1}}{(2\pi)^{D-2} a^{D-3} \sqrt{k^2 + \lambda^2}}, \tag{35}$$

with $0 \leq \lambda < \infty$.

In the L-region, from the normalizability of the mode functions, it follows that in (29) $c_2 = 0$ for $D \geq 4$. For $D = 3$, an additional condition on the AdS boundary is required in order to uniquely define the modes. Here, we consider a special case with $c_2 = 0$ for $D = 3$ as well. For the mode functions, we obtain

$$A_{\sigma\mu}(x) = C_{(\mathrm{L})\sigma} \epsilon_{(\gamma)\mu} z^{D/2-1} J_{D/2-1}(\lambda z) e^{i\mathbf{kx} - i\omega t}. \tag{36}$$

The allowed values for $\lambda$ are determined by the boundary condition on the brane and they are the roots of the equation

$$J_\nu(\lambda z_0) = 0, \tag{37}$$

with $\nu$ from (33). Hence, we have $\lambda = \lambda_{\nu,n}/z_0$. For the normalization coefficient, one gets

$$C_{(L)\sigma}^2 = \frac{2(2\pi)^{2-D}a^{3-D}}{z_0\sqrt{k^2 z_0^2 + \lambda_{\nu,n}^2}J_\nu'^2(\lambda_{\nu,n})}. \tag{38}$$

The mode functions in the region between two branes have recently been considered [49].

### 2.4. Boundary Conditions in $Z_2$-Symmetric Braneworlds

An example of the $Z_2$-symmetric braneworld is provided by the Randall–Sundrum model with a single brane (RSII model) formulated in background of (4+1)-dimensional AdS spacetime (see [77,78] and the review [7] for the RSI and RSII models). For an arbitrary number of spatial dimensions, the line element is given by (1) with $e^{-2y/a}$ replaced by $e^{-2|y|/a}$. The regions $-\infty < y < 0$ and $0 < y < +\infty$ are identified by the $Z_2$-symmetry. The brane is located at $y = 0$. Hence, in the corresponding setup, two copies of the R-region are employed with $z_0 = a$. The boundary conditions on the bulk fields at the location of the brane are obtained by integrating the field equations about $y = 0$ (see, e.g., the discussions in [26,40,67,76,79,80]).

For scalar fields, even under the reflection with respect to the brane, the Robin boundary condition is obtained with the coefficient $\beta = -2/(c_b + 4D\xi/a)$, where $c_b$ is the brane mass term. The latter appears in the part of the action located on the brane, $S_b = -c_b \int d^D x dy \sqrt{|g|} \delta(y) \varphi^2/2$. For odd scalar fields, the Dirichlet boundary condition is obtained. For fermionic fields two types of boundary conditions are obtained. The first one is reduced to the bag boundary condition (21) and the second one is obtained from (21) by the change of the sign in the term containing the normal to the brane. For vector fields, even under the reflection with respect to the brane, the boundary condition is reduced to (32) and for odd fields the condition (31) is obtained. For fermionic fields with the boundary condition obtained from (21) by the change of the sign in the second term, the VEV of the energy–momentum tensor is evaluated in a way similar to that we have demonstrated for the bag boundary condition. The corresponding mode functions are obtained from (22) by the replacement $ma + 1/2 \to ma - 1/2$ in the first index of the functions $g_{\mu,\rho}(x,u)$, $g_{ma+1/2,ma+s/2}(\lambda z_0, \lambda z) \to g_{ma-1/2,ma+s/2}(\lambda z_0, \lambda z)$, and in the expression (24) for the normalization coefficient.

Hence, we conclude that the VEVs of the energy–momentum tensors for scalar, Dirac and electromagnetic fields in $Z_2$-symmetric braneworlds with a single brane are obtained from the results given below for the R-region by an appropriate choice of the boundary conditions on the brane. The only difference is that an additional factor $1/2$ should be added. The latter is related to the presence of two copies of the R-region.

## 3. Vacuum Energy—Momentum Tensor

### 3.1. General Properties

Having the mode functions for quantum fields, we can investigate the VEVs of the local characteristics of the vacuum state. Among the most important characteristics is the VEV of the energy–momentum tensor. In particular, it determines the distribution of the vacuum energy density and the forces acting on

the boundaries (the Casimir forces). For a free field $\Psi(x)$ (the only interaction is that with background gravitational field), the operator of the energy–momentum tensor is a bilinear form in the field operator:

$$T_{\mu\rho} = T_{\mu\rho}\left\{\Psi(x), \Psi(x)\right\}. \tag{39}$$

The corresponding expressions of the bilinear form in the right-hand side for scalar, Dirac and vector fields can be found, for example, in [81,82]. Expanding the field operator in terms of a complete set $(\Psi_\sigma^{(+)}(x), \Psi_\sigma^{(-)}(x))$ of the positive and negative energy mode functions, specified by quantum numbers $\sigma$, using the commutation relations for the creation and annihilation operators and the definition of the vacuum state $|0\rangle$, for the VEV of the energy–momentum tensor the following sum over the modes is obtained:

$$\langle 0| T_{\mu\rho} |0\rangle \equiv \langle T_{\mu\rho}\rangle = \frac{1}{2}\sum_\sigma \sum_{s=\pm} T_{\mu\rho}\left\{\Psi_\sigma^{(s)}(x), \Psi_\sigma^{(s)*}(x)\right\}. \tag{40}$$

Here, $\sum_\sigma$ is understood as summation for discrete components of $\sigma$ and as integration over the continuous components. The expression in the right-hand side of (40) is divergent and a regularization procedure is required. The regularization can be made by point splitting, by using the local zeta function technique or by introducing a cutoff function [81–86]. In the presence of boundaries, the VEV (40) is decomposed into the boundary-free and boundary-induced contributions. The structure of the divergences is uniquely determined by the local geometric characteristics of the background spacetime. For points away from boundaries, the local geometry in the problems without and with boundaries is the same and, hence, the divergences are the same as well. From here it follows that at those points the boundary-induced contributions in the VEVs of local observables are finite and the renormalization procedure is the same as that in the boundary-free geometry. Consequently, for points outside the boundaries, the renormalization in (40) is reduced to that for the boundary-free part and the regularization dependences may appear in that part only. For definiteness, we will assume that a cutoff function is introduced in (40) without writing it explicitly. In [47–49,61,62,67], the point-splitting regularization technique is used. The details of the evaluation for the boundary-induced contributions at points outside the boundaries do not depend on the specific regularization scheme.

In the geometry with a brane, we decompose the vacuum energy–momentum tensor into two contributions:

$$\langle T_{\mu\rho}\rangle = \langle T_{\mu\rho}\rangle_0 + \langle T_{\mu\rho}\rangle_{\mathrm{b}}, \tag{41}$$

where $\langle T_{\mu\rho}\rangle_0$ is the VEV in the absence of the brane and the part $\langle T_{\mu\rho}\rangle_{\mathrm{b}}$ is induced by the brane. The VEV in the brane-free AdS geometry has been widely discussed in the literature (for recent discussion see, for example, [87,88]) and here we are mainly interested in the brane-induced effects. From the maximal symmetry of the AdS spacetime it follows that $\langle T_{\mu\rho}\rangle_0 = \mathrm{const}\cdot g_{\mu\rho}$. On the basis of the symmetry of the problem with a brane parallel to the AdS boundary, we expect that the VEV $\langle T_{\mu\rho}\rangle_{\mathrm{b}}$ is diagonal. From the Lorentz invariance in the subspace $(t, x^1, \ldots, x^{D-1})$, one concludes that the stresses in the directions parallel to the brane are equal to the energy density:

$$\left\langle T_0^0\right\rangle_{\mathrm{b}} = \left\langle T_1^1\right\rangle_{\mathrm{b}} = \cdots = \left\langle T_{D-1}^{D-1}\right\rangle_{\mathrm{b}}. \tag{42}$$

An additional relation between the components of the brane-induced VEV is obtained from the covariant continuity equation $\nabla_\mu \left\langle T_\rho^\mu\right\rangle_{\mathrm{b}} = 0$. The latter is reduced to

$$z^{D+1}\partial_z \left(z^{-D}\left\langle T_D^D\right\rangle_{\mathrm{b}}\right) + D\left\langle T_0^0\right\rangle_{\mathrm{b}} = 0. \tag{43}$$

As shown below, the brane-induced contribution in the VEVs depend on the coordinate $z$ and on the location of the brane through the ratio $z/z_0$. This property is a consequence of the maximal symmetry of the AdS spacetime and of the vacuum state we consider here. In addition, for points outside the brane, the following trace relations take place for scalar, Dirac and electromagnetic fields:

$$
\begin{aligned}
\langle T_\mu^\mu \rangle_b^{(s)} &= \left[ D(\xi - \xi_D)\nabla_\mu \nabla^\mu + m^2 \right] \langle \varphi^2 \rangle_b, \\
\langle T_\mu^\mu \rangle_b^{(f)} &= m \langle \bar{\psi}\psi \rangle_b, \\
\langle T_\mu^\mu \rangle_b^{(v)} &= -\frac{D-3}{16\pi} \langle F_{\mu\rho}F^{\mu\rho} \rangle_b.
\end{aligned}
\tag{44}
$$

Here and below, we use the superscripts (s),(f),(v) for the brane-induced energy–momentum tensors in the cases of scalar, Dirac and electromagnetic fields. On the right-hand sides of (44) the subscript b stands for the brane-induced contributions in the corresponding VEVs. For conformally coupled fields (conformally coupled massless scalar field ($\xi = \xi_D$, $m = 0$), massless Dirac field, the electromagnetic field in $D = 3$ spatial dimensions) the brane-induced energy–momentum tensor is traceless. For points away from the brane the trace anomaly is contained in the part $\langle T_{\mu\rho} \rangle_0$ only (on trace anomalies for different fields see, for example, [82]). The procedure for the evaluation of the brane-induced contribution in the VEV of the energy–momentum tensor follows similar steps for scalar, fermion and electromagnetic fields and we illustrate it on the example of the fermionic field.

The geometry given by (2) is conformally flat and for conformally coupled fields the problem under consideration is conformally related to the corresponding problem in the Minkowski bulk. Denoting by $\left\langle T_\mu^\rho \right\rangle_b^{(M)}$ the boundary-induced VEV in the Minkowskian problem, we have the following relation

$$
\left\langle T_\mu^\rho \right\rangle_b = (z/a)^{D+1} \left\langle T_\mu^\rho \right\rangle_b^{(M)}.
\tag{45}
$$

For the R-region, the problem with a brane in AdS bulk parallel to the AdS boundary is conformally related to the problem in the Minkowski bulk with the line element $ds_M^2 = \eta_{\mu\rho}dx^\mu dx^\rho$, $x^D = z$, and with a single boundary at $z = z_0$. For the L-region, the Minkowskian counterpart contains two boundaries. The first one is located at $z = z_0$ and is the conformal image of the brane and the second one is located at $z = 0$ and is the conformal image of the AdS boundary. The boundary conditions on $z = 0$ for fields in the Minkowskian problem are related to the special boundary conditions we have imposed on the AdS boundary.

### 3.2. R-Region

We describe the procedure for the evaluation of the brane-induced energy–momentum tensor on the example of the Dirac field. The procedure for scalar and electromagnetic fields follows similar steps.

### 3.2.1. Dirac Field

By using the expression for the energy–momentum tensor of the Dirac field and the corresponding mode functions from the previous section, the diagonal components in the R-region are presented in the form

$$
\langle T_\mu^\rho \rangle^{(f)} = \frac{\delta_\mu^\rho N a^{-D-1} (z/z_0)^{D+2}}{2(4\pi)^{(D-1)/2}\Gamma((D-1)/2)} \int_0^\infty du\, u^{D-2}
$$
$$
\times \int_0^\infty dx \frac{x}{\sqrt{x^2+u^2}} \frac{f_{(R)}^{(\mu)}(x, xz/z_0)}{J_{ma+1/2}^2(x) + Y_{ma+1/2}^2(x)}, \tag{46}
$$

where

$$
f_{(R)}^{(0)}(x,y) = -\left(x^2+u^2\right)\left[g_{ma+1/2,ma+s/2}^2(x,y) + g_{ma+1/2,ma-s/2}^2(x,y)\right],
$$
$$
f_{(R)}^{(D)}(x,y) = x^2 \left[g_{ma+1/2,ma+s/2}^2(x,y) + g_{ma+1/2,ma-s/2}^2(x,y)\right.
$$
$$
\left. -\frac{2ma}{y} g_{ma+1/2,ma+s/2}(x,y) g_{ma+1/2,ma-s/2}(x,y)\right]. \tag{47}
$$

As seen from (47), the VEVs for $s=+1$ and $s=-1$ coincide. For the separation of the brane-induced part, we use the relation

$$
\frac{g_{\mu,\rho}(x,u) g_{\mu,\rho'}(x,u)}{J_\mu^2(x) + Y_\mu^2(x)} = J_\rho(u) J_{\rho'}(u) - \frac{1}{2}\sum_{s=1,2} \frac{J_\mu(x)}{H_\mu^{(s)}(x)} H_\rho^{(s)}(u) H_{\rho'}^{(s)}(u), \tag{48}
$$

where $H_\rho^{(s)}(u)$, $s=1,2$, are the Hankel functions. For the separate terms in the integrand of (46), one has $\mu = ma + 1/2$, $\rho, \rho' = ma \pm s/2$ and $u = xz/z_0$. The part in the VEV coming from the first term in the right-hand side of (48) is the vacuum energy–momentum tensor in the geometry where the brane is absent (the part $\langle T_{\mu\rho} \rangle_0$ in (41)). In the complex plane $x = re^{i\phi}$, with $r$ being the modulus of $x$, for $z > z_0$ and for large values of $r$ the $s=1$ term in (48) (with $u = xz/z_0$) is exponentially small in the quarter $0 < \phi \le \pi/2$ and the $s=2$ term is exponentially small in the quarter $-\pi/2 \le \phi < 0$. On the basis of these properties, in the parts of the energy–momentum tensor with the last term in (48), we rotate the integration contour over $x$ by the angle $\pi/2$ for $s=1$ terms and by the angle $-\pi/2$ for $s=-1$ terms. Introducing the modified Bessel functions $I_\mu(x)$ and $K_\mu(x)$, we get

$$
\langle T_\mu^\rho \rangle_b^{(f)} = -\frac{2^{-D}\delta_\mu^\rho N}{\pi^{D/2}\Gamma(D/2)a^{D+1}} \int_0^\infty dx\, x^{D+1} \frac{I_{ma+1/2}(xz_0/z)}{K_{ma+1/2}(xz_0/z)} F_{(R)}^{(\mu)}(x), \tag{49}
$$

with the notations

$$
F_{(R)}^{(0)}(x) = \frac{1}{D}\left[K_{ma+1/2}^2(x) - K_{ma-1/2}^2(x)\right],
$$
$$
F_{(R)}^{(D)}(x) = K_{ma-1/2}^2(x) - K_{ma+1/2}^2(x) + \frac{2ma}{x} K_{ma+1/2}(x) K_{ma-1/2}(x). \tag{50}
$$

For massive fields both the energy density $\langle T_0^0 \rangle_b^{(f)}$ and the normal stress $\langle T_D^D \rangle_b^{(f)}$ are negative. As seen from (49), the brane-induced contribution depends on the coordinates $z$ and $z_0$ through the ratio $z/z_0 = e^{(y-y_0)/a}$, where $y_0 = a\ln(z_0/a)$ is the location of the brane in terms of the coordinate $y$. Note that $y - y_0$ is the proper distance from the brane. For a massless field, the brane-induced contribution vanishes, $\langle T_\mu^\rho \rangle_b = 0$.

The massless fermionic field is conformally invariant and this result could be directly obtained from the corresponding result for a single boundary in the Minkowski bulk by using the relation (45).

The Minkowskian result for a massive field is obtained from (49) in the limit $a \to \infty$ for fixed $y$ (see (1)). From the relation $z = ae^{y/a}$, it follows that $z \approx a + y$ and the values for the coordinate $z$ are large. By using the uniform asymptotic expansions for the modified Bessel functions for large values of the order, the leading term in the expansion of the components with $\mu = 0, 1, \ldots, D - 1$ coincides with the VEV for a planar boundary in the Minkowski bulk:

$$\langle T^\rho_\mu \rangle^{(\mathrm{Mf})}_{\mathrm{b}} = -\frac{2^{-D}\delta^\rho_\mu Nm}{\pi^{D/2}D\Gamma(D/2)} \int_m^\infty dx \, (x^2 - m^2)^{D/2} \frac{e^{-2x(y-y_0)}}{x + m}, \tag{51}$$

and the normal stress vanishes, $\langle T^D_D \rangle^{(\mathrm{Mf})}_{\mathrm{b}} = 0$.

The general formula (49) for the brane-induced contribution in the energy–momentum tensor is simplified in the region near the brane and at large distances from it (near-horizon limit). For large values $x$, the integrand in (49) behaves as $x^{D-1}e^{-2x(1-z_0/z)}$, and the integral is divergent for points on the brane. These surface divergences are well-known in quantum field theory with boundaries and have been widely discussed in the literature for different geometries of boundaries (see, for instance, [8–11]). Near the brane, assuming that $z/z_0 - 1 \ll 1$, the contribution of large values of $x$ dominate in the integral, and, by making use of the corresponding asymptotic formulas for the modified Bessel functions, in the leading order, we obtain

$$\langle T^0_0 \rangle^{(\mathrm{f})}_{\mathrm{b}} \approx -\frac{Nm\Gamma((D+1)/2)}{(4\pi)^{(D+1)/2}D\,(y-y_0)^D}, \quad \langle T^D_D \rangle^{(\mathrm{f})}_{\mathrm{b}} \approx -\frac{DNm\Gamma((D-1)/2)}{2(4\pi)^{(D+1)/2}Da(y-y_0)^{D-1}}. \tag{52}$$

In terms of the coordinate $y$, these asymptotics are valid under the conditions $y - y_0 \ll a, m^{-1}$. The leading terms for the energy density and for the stresses parallel to the brane coincide with those for a boundary in the Minkowski bulk. This is related to the fact that near the brane the dominant contribution to the VEV comes from the vacuum fluctuations with wavelengths smaller than the curvatures radius and the influence of the gravitational field on those fluctuations is weak. The effects of gravity are essential at distances from the brane larger than the curvature radius, $z/z_0 \gg 1$ or $y - y_0 \gg a$. In that region, the leading term in the asymptotic expansion for the energy density is expressed as

$$\langle T^0_0 \rangle^{(\mathrm{f})}_{\mathrm{b}} \approx -Nm\frac{\exp[-(2m + 1/a)(y - y_0)]}{2^{D+2ma+1}\pi^{(D-1)/2}a^D}$$
$$\times \frac{\Gamma(ma + (D+3)/2)\Gamma(2ma + D/2 + 1)}{(2ma + 1)\Gamma^2(ma + 1/2)\Gamma(ma + D/2 + 2)}, \tag{53}$$

and for the normal stress we get $\langle T^D_D \rangle^{(\mathrm{f})}_{\mathrm{b}} \approx D\langle T^0_0 \rangle^{(\mathrm{f})}_{\mathrm{b}} / (D + 2ma + 1)$. Note that, for a boundary in the Minkowski bulk and for $y - y_0 \gg 1/m$, the VEV decays as $e^{-2m(y-y_0)}$. In the AdS spacetime, the decay of the boundary-induced contribution, as a function of the proper distance from the boundary, is stronger.

The left panel of Figure 1 presents the dependence of the brane-induced contributions to the energy density and the normal stress (in units of $1/a^{D+1}$), $\langle T^\mu_\mu \rangle^{(\mathrm{f})}_{\mathrm{b}}$ (no summation over $\mu$), $\mu = 0, D$, on the ratio $z/z_0$ for fixed value $ma = 1$. In the right panel, we display the brane-induced parts as functions of the field mass for fixed $z/z_0 = 2$. For both panels, the full and dashed curves correspond to $D = 3$ and $D = 4$, respectively.

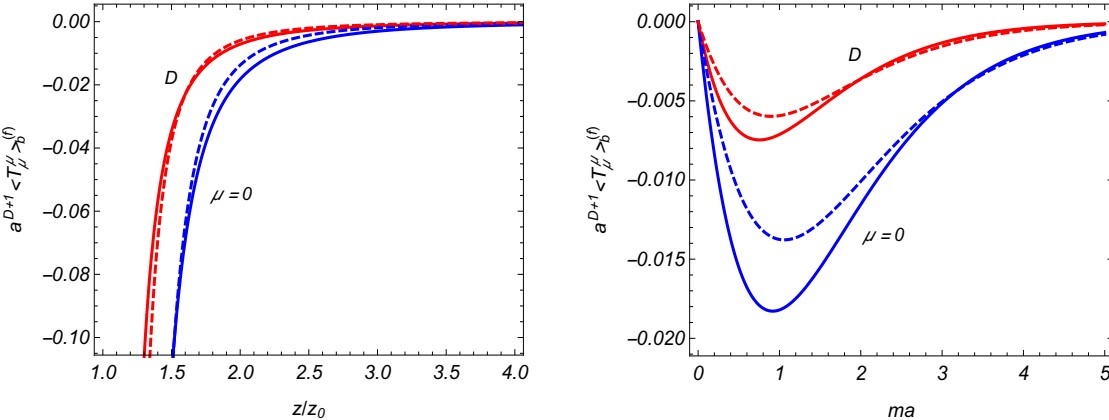

**Figure 1.** The brane-induced contributions in the VEVs of the energy density and the normal stress as functions: of $z/z_0$ (**left**); and of $ma$ (**right**). For the left panel, we take $ma = 1$ and for the right panel $z/z_0 = 2$. The full and dashed curves correspond to $D = 3$ and $D = 4$.

### 3.2.2. Scalar Field

In a similar way, for the brane-induced contribution in the case of a scalar field with the boundary condition (9), one obtains

$$\langle T_\mu^\rho \rangle_{\rm b}^{\rm (s)} = -\frac{2^{-D}\delta_\mu^\rho}{\pi^{D/2}\Gamma(D/2)a^{D+1}} \int_0^\infty dx\, x^{D+1} \frac{I_\nu(xz_0/z)}{\bar{K}_\nu(xz_0/z)} S^{(\mu)}\left[K_\nu(x)\right], \tag{54}$$

where the notation with the bar is defined as (10) and

$$
\begin{aligned}
S^{(0)}[g(x)] &= -\xi_1\left[g'^2(x) + \frac{D + 4\xi/\xi_1}{x}g(x)g'(x) + \left(1 + \frac{2}{D\xi_1} + \frac{\nu^2}{x^2}\right)g^2(x)\right], \\
S^{(D)}[g(x)] &= -g'^2(x) + \frac{D\xi_1}{x}g(x)g'(x) + \left(1 + \frac{2m^2a^2 - \nu^2}{x^2}\right)g^2(x),
\end{aligned}
\tag{55}
$$

with

$$\xi_1 = 4\xi - 1. \tag{56}$$

For a conformally coupled massless scalar field, the brane-induced contribution (54) vanishes. In the limit $a \to \infty$, with fixed $y$ and $y_0$, from (54), we get the vacuum energy–momentum tensor for a Robin boundary in the Minkowski bulk:

$$\langle T_\mu^\rho \rangle_{\rm b}^{\rm (Ms)} = \frac{2^{-D}\delta_\mu^\rho}{\pi^{D/2}\Gamma(D/2)} \int_m^\infty dx\, \left(x^2 - m^2\right)^{D/2-1} \left[\frac{m^2}{D} - 4\left(\xi - \xi_D\right)x^2\right] \frac{\beta x + 1}{\beta x - 1} e^{-2x(y - y_0)}, \tag{57}$$

for $\mu = 0, 1, 2, \ldots, D - 1$ and $\langle T_D^D \rangle_{\rm b}^{\rm (Ms)} = 0$.

For points near the brane, $y - y_0 \ll a, m^{-1}, |\beta|$, the leading term in the asymptotic expansion of the energy density is given by the expression

$$\langle T_0^0 \rangle_{\rm b}^{\rm (s)} \approx \pm \frac{D\Gamma\left((D+1)/2\right)\left(\xi - \xi_D\right)}{2^D \pi^{(D+1)/2}(y - y_0)^{D+1}}, \tag{58}$$

where the upper and lower signs correspond to $\beta = 0$ (Dirichlet boundary condition) and $\beta \neq 0$ (non-Dirichlet boundary conditions), respectively. The leading term in (58) coincides with that for the Minkowski bulk. For the normal stress we find $\langle T_D^D \rangle_b^{(s)} \approx \langle T_0^0 \rangle_b^{(s)} (y - y_0)/a$. In the case of a conformally coupled field, the leading terms vanish and the divergences on the brane are weaker. At large distances from the brane and for $\nu > 0$, the energy density is approximated as

$$
\begin{aligned}
\langle T_0^0 \rangle_b^{(s)} &\approx \frac{(2\nu - 1)e^{-2\nu(y-y_0)/a}}{2^{2\nu+D-1}\pi^{(D-1)/2}a^{D+1}} \frac{A_0 + B_0\nu}{A_0 - B_0\nu} \left( 4\xi - \frac{D + 2\nu}{D + 2\nu + 1} \right) \\
&\times \frac{\Gamma(D/2 + \nu + 1)\Gamma(D/2 + 2\nu)}{\Gamma(\nu + 1)\Gamma(D/2 + 1/2 + \nu)}.
\end{aligned}
\tag{59}
$$

For the normal stress, we get $\langle T_D^D \rangle_b^{(s)} \approx D\langle T_0^0 \rangle_b^{(s)}/(D + 2\nu)$. Note that for $\nu > 0$ the decay of the brane-induced VEV is exponential for both massless and massive fields. In the Minkowski bulk, the boundary-induced contribution decays as $e^{-2m(y-y_0)}$ for massive fields and as power-law $(y - y_0)^{-D-1}$ for non-conformally coupled massless fields.

In the left panel of Figure 2, we display the brane-induced energy density in the R-region as a function of $z/z_0$ for $D = 4$ minimally coupled scalar field with $ma = 1$. The graphs are plotted for the Dirichlet and Neumann boundary conditions and for the Robin boundary conditions with $\beta/a = -0.1, -0.15, -0.5$. The graphs for the Robin boundary conditions are located between the curves corresponding to the Dirichlet and Neumann conditions and for them $\langle T_0^0 \rangle_b^{(s)}$ increases with increasing $|\beta|$. The normal stress displays a similar behavior as a function of $z/z_0$. The right panel in Figure 2 presents the brane-induced contributions to the energy density ($\mu = 0$, full curves) and normal stress ($\mu = D$, dashed curves) for fixed $z/z_0 = 2$ as functions of $ma$ for $D = 4$ minimally coupled field. The graphs are plotted for the Dirichlet and Neumann boundary conditions and for the Robin boundary condition with $\beta/a = -0.15$. In the latter case, the energy density changes the sign at $ma \approx 3.13$ and the normal stress changes the sign at $ma \approx 3.34$.

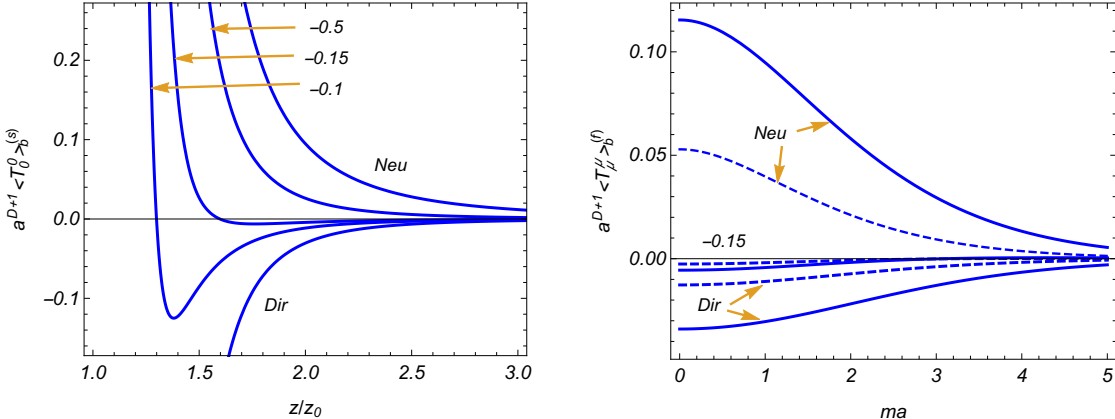

**Figure 2.** The brane-induced contributions in the VEVs of the energy density and the normal stress for $D = 4$ minimally coupled scalar field and for: (**left**) $ma = 1$; and (**right**) $z/z_0 = 2$. The full and dashed curves on the right panel correspond to the energy density and normal stress, respectively (for the boundary conditions chosen, see the text).

Figure 3 presents similar results for $D = 4$ conformally coupled field ($\xi = 3/16$). Note that in this case the stability condition $a/\beta < \nu - D/2$ on the Robin coefficient is stronger. For the left panel, we take $ma = 2$. With this value, for all the boundary conditions with $\beta \leq 0$, the vacuum is stable. The right panel

is plotted for $z/z_0 = 2$. There is a range for the values of $ma$ in which the Neumann boundary condition for a conformally coupled field leads to the vacuum instability.

An interesting feature that is seen from the graphs is the sign change and nonmonotonicity of the brane-induced energy density (as a function of the distance from the brane) for the Robin boundary condition. For a boundary in the Minkowski bulk, such behavior is easily obtained from the integral representation (57). Indeed, for points near the boundary, the dominant contribution to the integral in (57) comes from large values of $x$, and for $\beta \neq 0$ the leading term coincides with that for the Neumann boundary condition. It is given by (58) with the lower sign. At large distances from the boundary, the contribution of the integration range near the lower limit dominates and to the leading order one gets

$$\langle T_0^0 \rangle_b^{(\text{Ms})} \approx \frac{(1-4\xi)\, m^{D/2+1} e^{-2m(y-y_0)}}{2^{D+1}\pi^{D/2}(y-y_0)^{D/2}} \frac{\beta m + 1}{\beta m - 1}. \tag{60}$$

For a given curvature coupling parameter, depending on the Robin coefficient, the energy density corresponding to (60) can be either negative or positive. In particular, for a minimally coupled field, we see that the energy density is positive near the boundary for $\beta \neq 0$ and negative at large distances in the range $|\beta| < 1/m$. This means that the energy density changes the sign at some intermediate value of the distance from the boundary. Similar analysis can be provided for a brane in AdS bulk. Near the brane, the leading term in the asymptotic expansion is the same and the energy density is positive for non-Dirichlet boundary conditions. At large distances, the asymptotic of the brane-induced VEV is given by (59). For massive minimally and conformally coupled fields, one has $\nu > 1/2$, and the sign of the energy density in (59) is determined by the sign of the fraction containing the Robin coefficient. In particular, at large distances, the energy density is negative for $|a/\beta + D/2| > \nu$. Under this constraint and for non-Dirichlet boundary conditions, the brane-induced energy density is positive near the brane and negative at large distances. Examples of this kind of behavior are given in Figures 2 and 3.

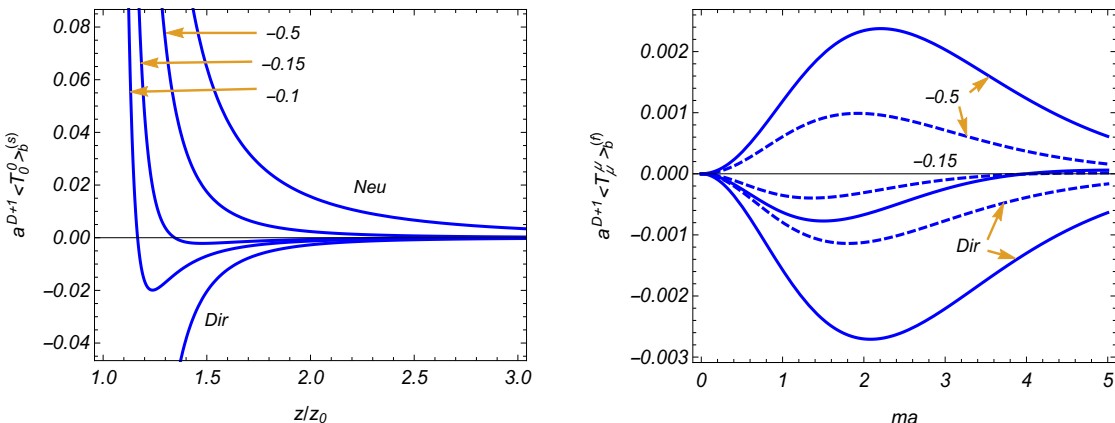

**Figure 3.** The same as in Figure 2 for $D = 4$ conformally coupled scalar field: (**left**) $ma = 2$; and (**right**) $z/z_0 = 2$.

### 3.2.3. Electromagnetic Field

For the electromagnetic field with the boundary conditions (31) and (32) on the brane, the brane-induced VEV has the form

$$\langle T_\mu^\rho \rangle_b^{(\text{v})} = -\frac{\delta_\nu \delta_\mu^\rho (D-1)}{2^D \pi^{D/2} \Gamma(D/2) a^{D+1}} \int_0^\infty dx\, x^{D+1} \frac{I_\nu(xz_0/z)}{K_\nu(xz_0/z)} V_{(\text{R})}^{(\mu)}(x), \tag{61}$$

where $\delta_\nu = 1$ for $\nu = D/2 - 1$ (boundary condition (31)), $\delta_\nu = -1$ for $\nu = D/2 - 2$ (boundary condition (32)), and

$$
\begin{aligned}
V_{(R)}^{(0)}(x) &= \left(1 - \frac{4}{D}\right) K_{D/2-1}^2(x) + \left(1 - \frac{2}{D}\right) K_{D/2-2}^2(x), \\
V_{(R)}^{(D)}(x) &= K_{D/2-1}^2(x) - K_{D/2-2}^2(x).
\end{aligned}
\tag{62}
$$

Note that the VEV has different signs for the boundary conditions (31) and (32). For $D = 3$, the electromagnetic field is conformally invariant and the VEV (61) is zero. One has (no summation over $\mu$) $\langle T_\mu^\mu \rangle_b^{(v)} < 0$ for the boundary condition (31) and $\langle T_\mu^\mu \rangle_b^{(v)} > 0$ for the condition (31). The VEVs for a plate in the Minkowski bulk are obtained from (61) in the limit $a \to \infty$:

$$
\langle T_\mu^\rho \rangle_b^{(Mv)} = \mp \delta_\mu^\rho \frac{(D-1)(D-3)\Gamma((D+1)/2)}{2\,(4\pi)^{(D+1)/2}\,(y-y_0)^{D+1}},
\tag{63}
$$

for $\mu = 0, 1, 2, \ldots, D-1$ and $\langle T_D^D \rangle_b^{(Mv)} = 0$. The upper and lower signs in (63) correspond to the conditions (31) and (32), respectively.

For $D > 3$, the VEV (61) diverges on the brane. The leading terms in the asymptotic expansions for the energy density and for the normal stress over the distance from the brane are expressed as

$$
\langle T_0^0 \rangle_b^{(v)} \approx -\delta_\nu \frac{(D-1)(D-3)\,\Gamma((D+1)/2)}{2\,(4\pi)^{(D+1)/2}\,(y-y_0)^{D+1}}, \quad \langle T_D^D \rangle_b^{(v)} \approx \frac{y-y_j}{a} \langle T_0^0 \rangle.
\tag{64}
$$

The leading term for the energy density coincides with the exact result for a boundary in the Minkowski bulk, given by (63). At large distances from the brane, assuming that $z_0/z \ll 1$, for $\nu > 0$ to the leading order, one gets

$$
\langle T_\mu^\rho \rangle_b^{(v)} \approx -\delta_\mu^\rho \frac{2^{1-D-2\nu}\delta_\nu\,(D-1)\,e^{-2\nu(y-y_0)}}{\pi^{D/2}\Gamma(D/2)\nu\Gamma^2(\nu)a^{D+1}} \int_0^\infty dx\, x^{D+2\nu+1} V_{(R)}^{(\mu)}(x),
\tag{65}
$$

where the integral is expressed in terms of the product of the gamma functions (see [89]). For $D = 4$ and the boundary condition (32), one has $\nu > 0$ and at large distances the brane-induced part in the energy–momentum tensor falls as $1/(y-y_0)$.

In Figure 4, we present the brane-induced energy density and the normal stress versus $z/z_0$ for $D = 4$ (left panel) and $D = 5$ (right panel) electromagnetic fields. The full and dashed curves correspond to the boundary conditions (31) and (32), respectively, and near the curves the value of the index $\mu$ is given for $\langle T_\mu^\mu \rangle_b^{(v)}$ (no summation over $\mu$) from (61).

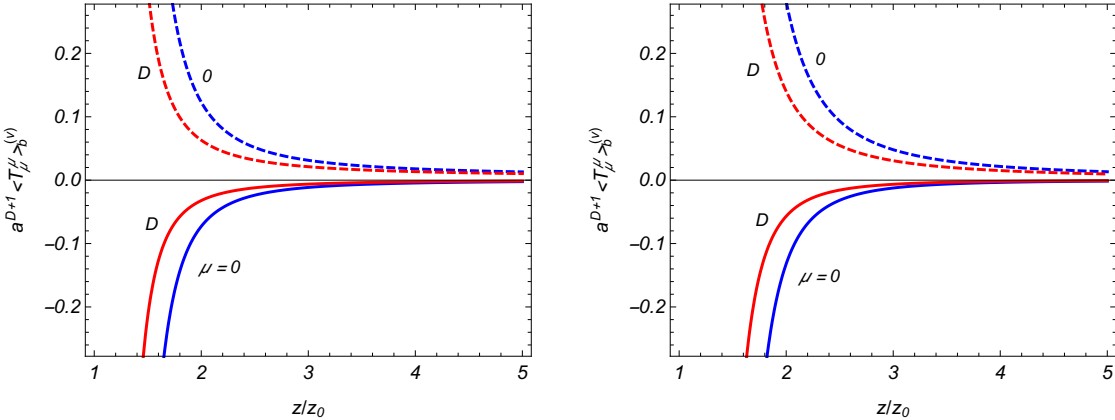

**Figure 4.** The brane-induced VEVs of the energy density and the normal stress as functions of $z/z_0$ for: $D = 4$ electromagnetic field (**left**); and $D = 5$ electromagnetic field (**right**). The full and dashed curves correspond to the conditions (31) and (32).

### 3.3. L-Region

In the L-region, the eigenvalues for the quantum number $\lambda$ are discrete. Again, we illustrate the evaluation procedure for the example of the Dirac field. The VEVs for scalar and electromagnetic fields are evaluated in a similar way.

#### 3.3.1. Dirac Field

By using the mode functions (25), the mode sum for the energy–momentum tensor is presented in the form

$$\langle T^\rho_\mu \rangle^{(\mathrm{f})} = -\frac{\delta^\rho_\mu (4\pi)^{(1-D)/2} N z^D}{\Gamma\left((D-1)/2\right) a^{D+1} z_0} \int_0^\infty dk\, k^{D-2} \sum_{n=1}^\infty \frac{f^{(\mu)}_{(\mathrm{L})}(\lambda_{ma-1/2,n} z/z_0)}{\sqrt{k^2 z_0^2 + \lambda^2_{ma-1/2,n}} J^2_{ma+1/2}(\lambda_{ma-1/2,n})}, \tag{66}$$

with the functions

$$\begin{aligned}
f^{(0)}_{(\mathrm{L})}(x) &= -(k^2 z^2 + x^2)\left[ J^2_{ma+s/2}(x) + J^2_{ma-s/2}(x) \right], \\
f^{(D)}_{(\mathrm{L})}(x) &= x^2 \left[ J^2_{ma+s/2}(x) + J^2_{ma-s/2}(x) - \frac{2ma}{x} J_{ma+s/2}(x) J_{ma-s/2}(x) \right].
\end{aligned} \tag{67}$$

By taking into account that the eigenvalues $\lambda_{ma-1/2,n}$ do not depend on $s$, we conclude that, similar to the R-region, the VEV of the energy–momentum tensor is the same for both inequivalent representations of the Clifford algebra.

For the summation of the series over the roots $\lambda_{ma-1/2,n}$, we use the Abel–Plana type formul [90,91]

$$\begin{aligned}
\sum_{n=1}^\infty \frac{f(\lambda_{ma-1/2,n})/\lambda_{ma-1/2,n}}{J^2_{ma+1/2}(\lambda_{ma-1/2,n})} &= \frac{1}{2}\int_0^\infty dx\, f(x) - \frac{1}{2\pi}\int_0^\infty dx\, \frac{K_{ma-1/2}(x)}{I_{ma-1/2}(x)} \\
&\quad \times \left[ e^{(1/2-ma)\pi i} f(ix) + e^{(ma-1/2)\pi i} f(-ix) \right].
\end{aligned} \tag{68}$$

The part of the VEV corresponding to the first term on the right-hand side of (68) gives the VEV in the geometry without the brane. For the brane-induced contribution coming from the second integral in (68), we get

$$\langle T_\mu^\rho \rangle_b^{(f)} = -\frac{2^{-D}\delta_\mu^\rho N}{\pi^{D/2}\Gamma(D/2)a^{D+1}} \int_0^\infty dx\, x^{D+1} \frac{K_{ma-1/2}(xz_0/z)}{I_{ma-1/2}(xz_0/z)} F_{(L)}^{(\mu)}(x), \tag{69}$$

where the functions in the integrand are defined as

$$
\begin{aligned}
F_{(L)}^{(0)}(x) &= \frac{1}{D}\left[I_{ma-1/2}^2(x) - I_{ma+1/2}^2(x)\right], \\
F_{(L)}^{(D)}(x) &= I_{ma+1/2}^2(x) - I_{ma-1/2}^2(x) + \frac{2ma}{x}I_{ma+1/2}(x)I_{ma-1/2}(x).
\end{aligned}
\tag{70}
$$

The energy density and the parallel stresses corresponding to (69) are negative, $\langle T_\mu^\mu \rangle_b^{(f)} < 0$ (no summation over $\mu$), $\mu = 0, 1, \ldots, D-1$, whereas the normal stress is positive. Comparing with the results for the R-region, we see that the energy densities in the R- and L-regions have the same sign and the normal stresses have opposite signs.

For a massless field one obtains $\langle T_D^D \rangle_b^{(f)} = -D\langle T_0^0 \rangle_b^{(f)}$ and the expression for the energy density is simplified to

$$\langle T_0^0 \rangle_b^{(f)} = -\left(\frac{z}{az_0}\right)^{D+1} \frac{N\Gamma((D+1)/2)}{(4\pi)^{(D+1)/2}}(1 - 2^{-D})\zeta(D+1), \tag{71}$$

where $\zeta(x)$ is the Riemann zeta function. In this case the brane-induced part is traceless. The massless fermionic field is conformally invariant and (71) corresponds to the conformal relation (45). In the Minkowskian counterpart, $z_0$ is the separation between two parallel planar boundaries. For a massive field and in the Minkowskian limit $a \to 0$ we obtain the expression (51) for the energy density and parallel stresses, with the replacement $y - y_0 \to y_0 - y$, and the normal stress vanishes.

The VEV (69) depends on $z$ and $z_0$ in the form of the ration $z/z_0$. Let us consider its behavior in the asymptotic regions. The limit $z/z_0 \to 1$ corresponds to the points on the brane. The brane-induced contribution (69) diverges on the brane. The leading terms in the asymptotic expansions over the distance from the brane are obtained from the corresponding expressions (52) for the R-region by the replacement $y - y_0 \to y_0 - y$ and by an additional change of the sign for the normal stress. For $z/z_0 \ll 1$ we use the small argument asymptotics for the functions $F_{(L)}^{(\mu)}(x)$. To the leading order this gives

$$\langle T_0^0 \rangle_b^{(f)} \approx -\frac{2^{-D-2ma}\pi^{-D/2}N(z/z_0)^{D+2ma+1}}{\Gamma(D/2+1)\Gamma^2(ma+1/2)a^{D+1}} \int_0^\infty dx\, x^{D+2ma} \frac{K_{ma-1/2}(x)}{I_{ma-1/2}(x)},$$

for the energy density and

$$\langle T_D^D \rangle_b^{(f)} \approx -\frac{D}{2ma+1}\langle T_0^0 \rangle_b^{(f)},$$

for the normal stress. In particular, the brane-induced VEV tends to zero on the AdS boundary as $z^{D+2ma+1}$.

The brane-induced contributions for the energy density and the normal stress in the L-region, $\langle T_\mu^\mu \rangle_b^{(f)}$ (no summation over $\mu$), $\mu = 0, D$, for $D = 3$ (full curves) and $D = 4$ (dashed curves) Dirac fields are presented in Figure 5 as functions of $z/z_0$ and $ma$. The left and right panels are plotted for $ma = 1$ and $z/z_0 = 0.6$, respectively.

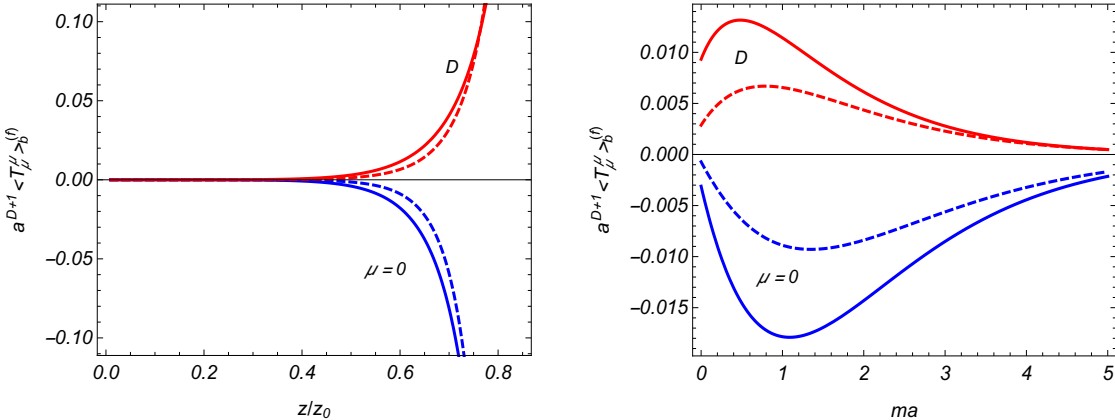

**Figure 5.** The same as in Figure 1 for the L-region. The left and right panels are plotted for $ma = 1$ and $z/z_0 = 0.6$, respectively.

#### 3.3.2. Scalar Field

For a scalar field, the mode-sum formula for the VEV of the energy–momentum tensor contains series over the zeros of the function $\bar{J}_\nu(x) = B_0 x J'_\nu(x) + A_0 J_\nu(x)$. The corresponding summation formula can be found in [90,91]. The brane-induced contribution for a scalar field is presented as

$$\langle T^\rho_\mu \rangle^{(s)}_b = -\frac{2^{-D}\delta^\rho_\mu}{\pi^{D/2}\Gamma(D/2)a^{D+1}} \int_0^\infty dx\, x^{D+1} \frac{\bar{K}_\nu(xz_0/z)}{\bar{I}_\nu(xz_0/z)} S^{(\mu)}\left[I_\nu(x)\right], \tag{72}$$

where the functions $S^{(\mu)}[g(x)]$ are given by (55). For a conformally coupled massless field, one has $\nu = 1/2$ and $S^{(0)}[I_{1/2}(x)] = 2/(\pi D x)$, $S^{(D)}[I_{1/2}(x)] = -2/(\pi x)$. For the energy density, we get

$$\langle T^0_0 \rangle^{(s)}_b = -\frac{(z/a)^{D+1}z_0^{-D-1}}{2^{2D+1}\pi^{D/2}\Gamma(D/2+1)} \int_0^\infty dx \frac{x^D}{\frac{2a/\beta+1-D-x}{2a/\beta+1-D+x}e^x - 1}, \tag{73}$$

and for the normal stress $\langle T^D_D \rangle^{(s)}_b = -D\langle T^0_0 \rangle^{(s)}_b$. Note that one has the conformal relation $\varphi(x) = (z/a)^{(D-1)/2}\varphi_M(x)$ between the fields in the AdS and Minkowski bulk. From here, it follows that the conformal image of the boundary condition (9) is the condition $(\beta_M n^\mu_M \nabla_\mu + 1)\varphi_M(x) = 0$, $z = z_0$, with the relation between the Robin coefficients

$$\frac{z_0}{\beta_M} = \frac{a}{\beta} - \frac{D-1}{2}. \tag{74}$$

By taking into account (74), we can see that (73) corresponds to the conformal relation (45) with the Minkowskian problem of two boundaries with the Dirichlet boundary condition on the plate $z = 0$ (conformal image of the AdS boundary) and the Robin condition with the coefficient $\beta_M$ on the plate at $z = z_0$. In the Minkowskian limit the result (57) is obtained for the energy density and parallel stresses and the normal stress becomes zero.

Let us consider the behavior of the brane-induced VEV (72) near the brane and near the AdS boundary. Near the brane, one has $1 - z/z_0 \ll 1$ and the contribution of large $x$ dominates in the integral of (72). For non-conformally coupled fields, the leading term in the expansion of the energy density is given by (58) with the replacement $y - y_0 \to y_0 - y$ and the relation between the energy density and the normal stress remain the same. In particular, we see that near the brane the energy density has the same sign in

the R- and L-regions, whereas the normal stresses have opposite signs. For points near the AdS boundary, one has $z/z_0 \ll 1$ and the leading term in the asymptotic expansion is given by

$$\langle T_0^0 \rangle_b^{(s)} \approx \frac{(z/z_0)^{D+2\nu}}{\pi^{D/2}\Gamma(D/2)} \frac{(D+2\nu+1)\,\xi_1+1}{2^{D+2\nu}\Gamma(\nu)\Gamma(\nu+1)a^{D+1}} \int_0^\infty dx x^{D+2\nu-1} \frac{\bar{K}_\nu(x)}{\bar{I}_\nu(x)}. \tag{75}$$

In the same order, the normal stress is found from $\langle T_D^D \rangle_b^{(s)} \approx -D\langle T_0^0 \rangle_b^{(s)}/(2\nu)$. Hence, the brane-induced VEVs vanish on the AdS boundary as $z^{D+2\nu}$. Note that the factor $(D+2\nu+1)\,\xi_1+1$ is negative for both minimally and conformally coupled field and for the Dirichlet boundary condition the energy density is negative near the AdS boundary. In general, depending on the Robin coefficient, the energy density can be either negative or positive.

The brane-induced energy density for $D=4$ minimally coupled scalar field is presented in the left panel of Figure 6 as a function of $z/z_0$. The graphs are plotted for $ma=1$, for the Dirichlet and Neumann boundary conditions (the labels Dir and Neu near the curves), and for the Robin boundary conditions with $\beta/a = -0.5, -0.15, -0.1$. For the latter cases, the energy density $\langle T_0^0 \rangle_b^{(s)}$ increases with increasing $|\beta/a|$. In the right panel, we display the energy density and the normal stress as functions of the mass for the Dirichlet and Neumann boundary conditions, and for the Robin condition with $\beta/a = -0.15$. The graphs are plotted for $z/z_0 = 0.6$. For the Robin boundary condition the brane-induced VEVs may change the sign as functions of the mass.

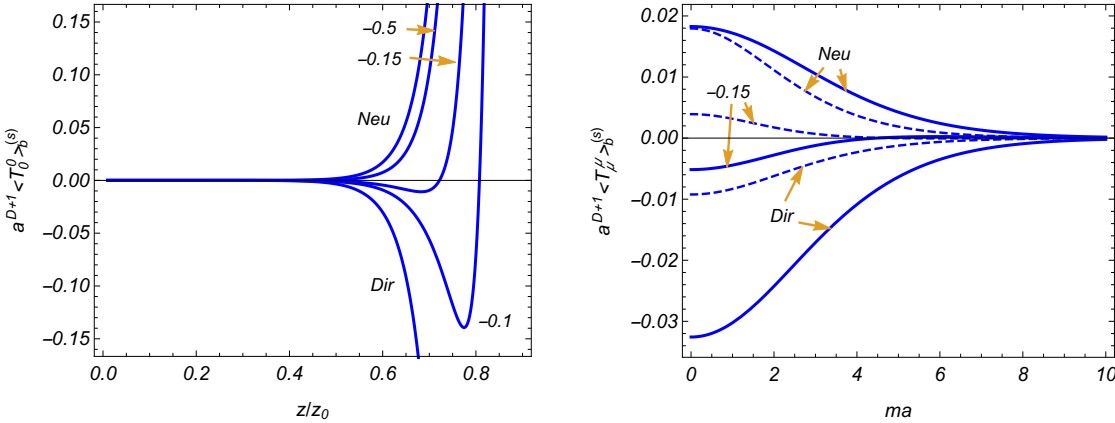

**Figure 6.** The same as in Figure 2 for a minimally coupled scalar field in the L-region. The left and right panels are plotted for $ma = 1$ and $z/z_0 = 0.6$, respectively.

It is of interest to compare the boundary-induced VEVs in AdS and dS spacetimes. The scalar Casimir densities in background of dS spacetime are investigated in [92,93] for a single and two parallel Robin boundaries, respectively. The corresponding line element is taken in planar coordinates,

$$ds^2 = dt^2 - e^{2t/a} \sum_{i=1}^D (dz^i)^2, \tag{76}$$

with $-\infty < t, z^i < +\infty$, and it is assumed that the field is prepared in the Bunch-Davies vacuum state. In the geometry of a single boundary at $z^D = 0$, the boundary-induced VEV of the energy–momentum tensor is expressed in terms of integrals that contain the products of the modified Bessel functions with the order $\nu_{dS} = \left[D^2/4 - D(D+1)\xi - m^2a^2\right]^{1/2}$ (compare with (7)). The VEV, in addition to

the diagonal components, has also nonzero off-diagonal component $\langle T_0^D \rangle_{\mathrm{b}}^{(\mathrm{s})}$ that describes the energy flux along the direction normal to the boundary. Depending on the boundary condition, the flux can be either positive or negative. Another qualitatively new effect of the gravity, compared with the corresponding problem in the Minkowski bulk, is the appearance of the nonzero normal stress $\langle T_D^D \rangle_{\mathrm{b}}^{(\mathrm{s})}$. All the components $\langle T_\mu^\rho \rangle_{\mathrm{b}}^{(\mathrm{s})}$ depend on the spacetime coordinates $t$ and $z^D$ in the form of the combination $|z^D| e^{t/a}$. The latter is the proper distance from the boundary measured in units of the curvature radius $a$. Similar to the case of the AdS bulk, for small values of that combination, the influence of the gravitational field on the energy density and stresses parallel to the boundary is weak. At large proper distances from the boundary, $|z^D| e^{t/a} \gg a$, the decay of the boundary-induced VEV is qualitatively different for real and purely imaginary values of $\nu_{\mathrm{dS}}$. For $\nu_{\mathrm{dS}} > 0$, the VEV tends to zero monotonically, as $(|z^D| e^{t/a})^{2\nu_{\mathrm{dS}} - D - \chi}$, where $\chi = 0$ for the diagonal components and $\chi = 1$ for the component $\langle T_0^D \rangle_{\mathrm{b}}^{(\mathrm{s})}$. For imaginary $\nu_{\mathrm{dS}}$, the behavior of the boundary-induced VEV at large proper distances is damping oscillatory, as $(|z^D| e^{t/a})^{-D-\chi} \sin \left[ 2|\nu_{\mathrm{dS}}| \ln(|z^D| e^{t/a}) + \phi \right]$, with $\phi$ being a constant phase. Recall that for the AdS bulk the decay of the boundary-induced VEV of the energy–momentum tensor, as a function of the proper distance $|y - y_0|$, was exponential at large distances (see (59) and (75)).

### 3.3.3. Electromagnetic Field

For the electromagnetic field, the eigenmodes of the quantum number $\lambda$ are roots of the Equation (37), where $\nu$ is given by (33) for the boundary conditions (31) and (32). The corresponding summation formula for the series in the mode-sum of the energy–momentum tensor is obtained from (68) by the replacement $ma - 1/2 \to \nu$. The brane-induced contribution is expressed as

$$\langle T_\mu^\rho \rangle_{\mathrm{b}}^{(\mathrm{v})} = -\frac{\delta_\nu \delta_\mu^\rho (D-1)}{2^D \pi^{D/2} \Gamma(D/2) a^{D+1}} \int_0^\infty dx \, x^{D+1} \frac{K_\nu(xz_0/z)}{I_\nu(xz_0/z)} V_{(\mathrm{L})}^{(\mu)}(x), \tag{77}$$

where we have defined the functions

$$\begin{aligned}
V_{(\mathrm{L})}^{(0)}(x) &= \left(1 - \frac{4}{D}\right) I_{D/2-1}^2(x) + \left(1 - \frac{2}{D}\right) I_{D/2-2}^2(x), \\
V_{(\mathrm{L})}^{(D)}(x) &= I_{D/2-1}^2(x) - I_{D/2-2}^2(x).
\end{aligned} \tag{78}$$

For $D \geq 4$, the energy density is negative for the boundary condition (31) and positive for the condition (32). For the normal stress, one has $\langle T_D^D \rangle_{\mathrm{b}}^{(\mathrm{v})} > 0$ in the case of the condition (31) and $\langle T_D^D \rangle_{\mathrm{b}}^{(\mathrm{v})} < 0$ for (32). In the Minkowskian limit $a \to \infty$, with fixed $y, y_0$, we obtain the result (63).

Unlike the R-region, in the L-region, the brane-induced contribution is different from zero for $D = 3$. For the boundary condition (31) one gets

$$\langle T_\mu^\rho \rangle_{\mathrm{b}}^{(\mathrm{v})} = -\frac{\pi^2}{720} \left(\frac{z}{az_0}\right)^4 \mathrm{diag}(1,1,1,-3). \tag{79}$$

This determines the Casimir force per unit surface of the brane equal to $-\pi^2 a^{-4}/240$. The latter is attractive with respect to the AdS boundary and does not depend on the location of the brane. In the case of the boundary condition (32), we obtain the expression

$$\langle T_\mu^\rho \rangle_{\mathrm{b}}^{(\mathrm{v})} = \frac{7\pi^2}{5760} \left(\frac{z}{az_0}\right)^4 \mathrm{diag}(1,1,1,-3). \tag{80}$$

The corresponding force per unit surface of the brane is given by $7\pi^2\alpha^{-4}/1920$ and is repulsive with respect to the AdS boundary.

The near-brane asymptotics in the L-region are given by (64) with the replacement $y - y_0 \to y_0 - y$ in the expression for the energy density. Near the AdS boundary, one has $z/z_0 \ll 1$, and, by using the expression of the modified Bessel function for small arguments, in the leading order, we get

$$\langle T_0^0 \rangle_b^{(v)} \approx -\frac{8\delta_\nu (D-1)(z/2z_0)^{2D-2}}{\pi^{D/2}D\Gamma^3(D/2-1)a^{D+1}} \int_0^\infty dx\, x^{2D-3}\frac{K_\nu(x)}{I_\nu(x)},$$

$$\langle T_D^D \rangle_b^{(v)} \approx -\frac{D}{D-2}\langle T_0^0 \rangle_b^{(v)}. \tag{81}$$

In spatial dimension $D = 3$, the integral is equal to $\pi^5/240$ for the boundary condition (31) and $7\pi^5/1920$ for (32). In this special case, the asymptotics (81) coincide with the exact results (79) and (80).

For $D = 4$ and $D = 5$ electromagnetic fields, the brane contributions to the vacuum energy density and the normal stress are plotted in the left and right panels of Figure 7 (the values of the index $\mu$ for $\langle T_\mu^\mu \rangle_b^{(v)}$ (no summation over $\mu$) are displayed near the curves), respectively. The full and dashed curves correspond to the boundary conditions (31) and (32), respectively.

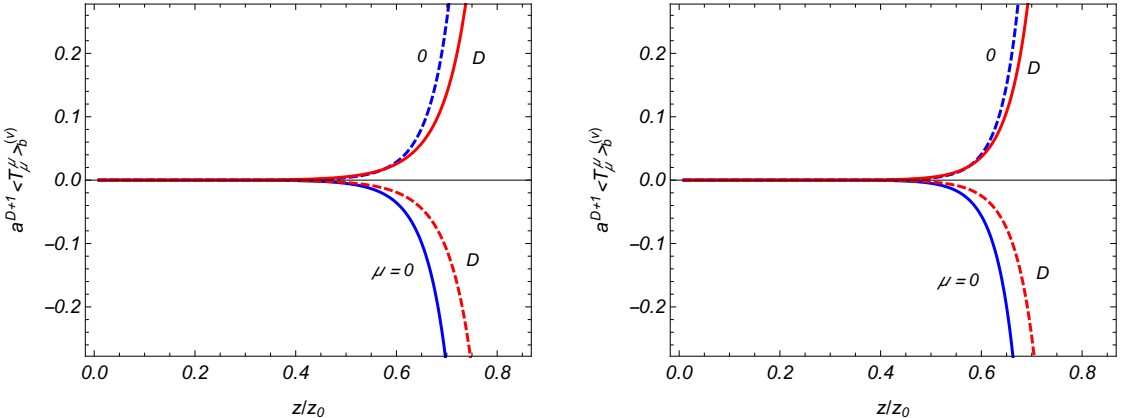

**Figure 7.** The brane-induced VEVs of the energy density and the normal stress in the L-region as functions of $z/z_0$ for: $D = 4$ electromagnetic field (**left**); and $D = 5$ electromagnetic field (**right**). The full and dashed curves correspond to the conditions (31) and (32).

The electromagnetic Casimir densities for the boundary condition (31) on a single and two parallel plates in dS spacetime with the line element (76) have been investigated in [94,95]. It was assumed that the field is prepared in the Bunch-Davies vacuum. For $D = 3$, the electromagnetic field is conformally invariant and the plate-induced contribution in the VEV of the energy–momentum tensor is conformally related to that for the Minkowski bulk. In particular, it vanishes in the geometry of a single plate. In spatial dimensions $D \geq 4$, the components of the plate-induced VEV are expressed in terms of the hypergeometric function. For these dimensions, similar to the case of a scalar field, the VEV of the energy–momentum tensor has nonzero off-diagonal component $\langle T_0^D \rangle_b^{(v)}$. For a single plate, the corresponding energy flux is directed from the plate. At large proper distances from the plate, located at $z^D = 0$, the vacuum stresses are isotropic and for $D > 4$ the diagonal components decay as $\left(|z^D|e^{t/a}\right)^{-4}$ and the off-diagonal component behaves like $(|z^D|e^{t/a})^{-5}$. For $D = 4$ the asymptotics for $\langle T_0^0 \rangle_b$ and $\langle T_0^D \rangle_b$ remain the same and the stresses behave as $(|z^D|e^{t/a})^{-6} \ln\left(|z^D|e^{t/a}\right)$. In dS spacetime, we have a power-law decay of the

boundary-induced VEV as a function of the proper distance from the plate. In the AdS bulk, the decay is exponential.

## 4. VEV of the Surface Energy—Momentum Tensor for a Scalar Field

In the discussion above, we consider the VEV of the bulk energy–momentum tensor. In manifolds with boundaries, in addition to the latter, a surface energy–momentum tensor may present, which is localized on the boundaries. In the general case of bulk and boundary geometries, the expression of the surface energy–momentum tensor for a scalar field with general curvature coupling has been obtained in [96] by using the standard variational procedure. The VEV of the surface energy–momentum tensor for branes parallel to the AdS boundary is investigated in [97] by using the generalized zeta function technique.

For a given field, the expression for the surface energy–momentum tensor $T_{\mu\rho}^{(\text{surf})}$, in addition to the bulk action, depends on the surface action. In [96], for a spacetime region $M$ with boundary $\partial M$, the surface action for a scalar field is taken in the form

$$S_s = -\epsilon \int_{\partial M} d^D x \sqrt{|h|} \, (\xi K + m_s) \, \varphi^2, \tag{82}$$

where $\epsilon = 1$ for spacelike and $\epsilon - 1$ for timelike elements of the boundary, and $h$ is the determinant of the induced metric $h_{\mu\rho} = g_{\mu\rho} - \epsilon n_\mu n_\rho$, with $n_\mu$ being the inward pointing unit normal to $\partial M$, $n_\mu n^\mu = \epsilon$. In (82), $K = g^{\mu\rho} K_{\mu\rho}$ is the trace of the extrinsic curvature tensor $K_{\mu\rho} = h_\mu^\sigma h_\rho^\tau \nabla_\sigma n_\tau$ of the boundary and $m_s$ is a parameter. $\partial M$ consists of the initial and final spacelike hypersurfaces and a timelike smooth boundary $\partial M_s$. The variation of the total action with respect to the field $\varphi(x)$ leads to the standard field Equation (3) in the bulk and to the boundary condition

$$\left(2\xi K + 2m_s + n^\mu \nabla_\mu\right) \varphi(x) = 0, \; x \in \partial M_s. \tag{83}$$

The variation of the action with respect to the metric tensor gives the metric energy–momentum tensor. In addition to the bulk part, the latter contains a contribution $T_{\mu\rho}^{(\text{surf})}$ located on the boundary $\partial M_s$: $T_{\mu\rho}^{(\text{surf})} = \tau_{\mu\rho} \delta(x; \partial M_s)$, where $\delta(x; \partial M_s)$ is the 'one-sided' $\delta$-function. By using the boundary condition (83), the expression for $\tau_{\mu\rho}$ is presented in the form [96]

$$\tau_{\mu\rho} = \xi \varphi^2 K_{\mu\rho} - \left(2\xi - \frac{1}{2}\right) h_{\mu\rho} \varphi n^\sigma \nabla_\sigma \varphi. \tag{84}$$

Note that the boundary condition (83) is of the Robin type. By using the boundary condition, one can exclude the derivative term for the field in (84).

In the geometry (2) with a single brane at $z = z_0$, the extrinsic curvature tensor for the R- and L-regions (J=R,L) has the form $K_{\mu\rho}^{(J)} = -\delta_{(J)} g_{\mu\rho}/a$ for $\mu, \rho = 0, 1, \ldots, D - 1$, and $K_{DD}^{(J)} = 0$. The boundary condition (83) is reduced to (9) with $1/\beta = 2m_s - 2\delta_{(J)} D\xi/a$. The VEV of the surface energy–momentum tensor, $\langle 0| \tau_{\mu\rho} |0\rangle \equiv \langle \tau_{\mu\rho} \rangle$, is evaluated by using the mode-sum formula $\langle \tau_{\mu\rho} \rangle = \sum_\sigma \sum_{s=\pm} \tau_{\mu\rho} \{\varphi_\sigma^{(s)}(x), \varphi_\sigma^{(s)*}(x)\}/2$ with the mode functions given by (12) and (15) for the R- and L-regions, respectively. The VEV has the form $\langle \tau_\mu^\rho \rangle = \text{const} \cdot \delta_\mu^\rho$, $\mu, \rho = 0, 1, \ldots, D - 1$, $\langle \tau_\mu^D \rangle = 0$ and, from the point of view of an observer living on the brane it corresponds to a gravitational source of the cosmological constant type. An essential difference compared with the bulk energy–momentum tensor is that the subtraction of the part corresponding to the geometry without a brane is not sufficient and an additional renormalization is required. The latter is reduced to the renormalization of the VEV for the field squared on the brane. In [97], the generalized zeta function technique is used.

The VEV of the surface energy–momentum tensor for the region J = R,L, $\langle \tau_{\mu\rho} \rangle^{(J)}$, is expressed in terms of the VEV of the field squared on the brane, $\langle \varphi^2 \rangle^{(J)}_{z=z_0}$, as

$$\left\langle \tau_{\mu}^{\rho} \right\rangle^{(J)} = \delta_{\mu}^{\rho} \delta_{(J)} \left[ (2\xi - 1/2)/\beta - \xi/a \right] \left\langle \varphi^2 \right\rangle^{(J)}_{z=z_0}, \tag{85}$$

The VEV $\left\langle \varphi^2 \right\rangle^{(J)}_{z=z_0}$ is obtained by the analytic continuation of the function (the details can be found in [97])

$$F_{(J)}(s) = -\frac{\delta_{(J)} (\sqrt{4\pi}a)^{1-D} \beta (\mu z_0)^{-s-1}}{\Gamma(-s/2)\Gamma((D+1+s)/2)} \int_0^{\infty} dx\, x^{D+s} U_{(J)\nu}(x), \tag{86}$$

to the physical point $s = -1$. Here, the parameter $\mu$ is the renormalization scale, $\nu$ is defined by (7) and

$$U_{(R)\nu}(x) = \frac{K_{\nu}(x)}{\bar{K}_{\nu}(x)}, \quad U_{(L)\nu}(x) = \frac{I_{\nu}(x)}{\bar{I}_{\nu}(x)}. \tag{87}$$

The representation (86) is valid in the slice $-(D+1) < \text{Re}\, s < -D$ of the complex $s$-plane. In the first step of the analytic continuation, the integral in (86) is presented in the form of the sum of the integrals over the regions $[0,1]$ and $[1,\infty)$. In the first integral the substitution $s = -1$ can be made directly. In the second integral, we subtract and add to the function $U_{(J)\nu}(x)$ in the integrand the $N$ leading terms of the corresponding asymptotic expansion for large values of $x$ and integrate the asymptotic part. For $\beta \neq 0$ the asymptotic expansion has the form $\beta U_{(J)\nu}(x) \sim \sum_{l=0}^{\infty} w_l(\nu)(-\delta_{(J)}x)^{-l-1}$, where the coefficients $w_l(\nu)$ are found from those for the expansions of the modified Bessel functions. The function $F_{(J)}(s)$ has a simple pole at $s = -1$ and the leading term in the Laurent expansion is given by

$$\frac{-2(-\delta_{(J)}a)^{1-D} w_{D-1}(\nu)}{(4\pi)^{D/2}\Gamma(D/2)(s+1)}. \tag{88}$$

In this way, the VEVs $\left\langle \varphi^2 \right\rangle^{(J)}_{z=z_0}$ for J = R and J = L are decomposed into the pole and finite contributions. The pole terms can be absorbed by adding to the brane action the respective counterterms. The expressions for the finite parts in separate regions will not be given here and can be found in [97]. We will consider the total energy density.

Combining the results for the R- and L-regions, one obtains the total surface energy density $\langle \tau_0^0 \rangle = \langle \tau_0^0 \rangle^{(R)} + \langle \tau_0^0 \rangle^{(L)}$. Comparing the pole parts (88) for $\left\langle \varphi^2 \right\rangle^{(J)}_{z=z_0}$ in those regions, we can see that in odd spatial dimensions the pole parts in the energy density cancel out and the finite part does not depend on the renormalization scale $\mu$. Taking $N = D - 1$ for the number of the terms taken in the asymptotic expansions of the function $U_{(J)\nu}(x)$, for the total surface energy density in odd dimensions $D$, one gets the formula

$$\left\langle \tau_0^0 \right\rangle = \frac{2\xi\beta/a + 1 - 4\xi}{(4\pi)^{D/2}\Gamma(D/2)a^D} \left[ \int_0^1 dx\, x^{D-1} \sum_{J=R,L} U_{(J)\nu}(x) - \frac{2}{\beta} \sum_{l=0}^{(D-3)/2} \frac{w_{2l+1}(\nu)}{D - 2l - 2} \right.$$
$$\left. + \int_1^{\infty} dx\, x^{D-1} \left( \sum_{J=R,L} U_{(J)\nu}(x) - \frac{2}{\beta} \sum_{l=0}^{(D-3)/2} \frac{w_{2l+1}(\nu)}{x^{2l+2}} \right) \right]. \tag{89}$$

Note that this quantity does not depend on the location of the brane. Depending on the value of the Robin coefficient $\beta$, the surface energy density (89) can be either positive or negative (see the graphs in [97] for minimally and conformally coupled scalar fields).

In the geometry of two branes, the VEV of the surface energy–momentum tensor on a given brane is decomposed into two parts. The first one corresponds to the VEV in the problem where the second brane is absent and the corresponding evaluation procedure has been described in this section. The second part is induced by the presence of the second brane and it requires no additional renormalization. As it has been discussed in [97], in the Randall–Sundrum model the surface energy density induced on the visible brane by the presence of the hidden brane gives rise to naturally suppressed cosmological constant. The surface energy density in models with additional compact dimensions is discussed in [98,99] for neutral and charged scalar fields. In the latter case, the value of the induced cosmological constant on the brane is additionally controlled by tuning the magnetic flux enclosed by compact dimensions.

## 5. Geometry with a Brane Perpendicular to the AdS Boundary

In a number of recent developments of the AdS/CFT correspondence, branes intersecting the AdS boundary are considered. They include the extensions of the correspondence for conformal field theories with boundaries (AdS/BCFT correspondence) [100,101] and the geometric procedure for the evaluation of the entanglement entropy for a bounded region in CFT [102,103]. In this section, based on [69], we consider the effects on the scalar vacuum induced by a brane perpendicular to the AdS boundary.

The background geometry is described by the line element (2) and the brane is located at $x^1 = 0$. The problem is symmetric with respect to the brane and we will consider the region $x^1 \geq 0$. The scalar field $\varphi(x)$ obeys the field Equation (3) and the boundary condition

$$(\beta \partial_1 + 1) \, \varphi(x) = 0, \tag{90}$$

on the brane. Introducing the notations $\mathbf{x} = (x^2, \ldots, x^{D-1})$ and $\mathbf{k} = (k_2, \ldots, k_{D-1})$, the normalized mode functions obeying the boundary condition have the form

$$\varphi_\sigma^{(\pm)}(x) = \frac{\sqrt{2k_1/\omega}}{(2\pi a)^{(D-1)/2}} z^{D/2} J_\nu(\lambda z) \cos[k_1 x^1 + \alpha_0(k_1)] e^{i\mathbf{k}\mathbf{x} \mp i\omega t}, \tag{91}$$

where $0 \leq k_1, \lambda < \infty$, the energy is given by $\omega = \sqrt{k^2 + k_1^2 + \lambda^2}$ with $k^2 = \sum_{i=2}^{D} k_i^2$ and $\nu$ is defined by (7). The function $\alpha_0(k_1)$ is determined from the relation

$$e^{2i\alpha_0(k_1)} = \frac{i\beta k_1 - 1}{i\beta k_1 + 1}. \tag{92}$$

Note that in the case $0 \leq \nu < 1$ we impose the same boundary condition on the AdS boundary as that in Section 2 for the L-region.

For positive values of the Robin parameter $\beta$ in (90), in addition to (91), there is a mode for which the part depending on the coordinate $x^1$ is expressed in terms of the exponential function $e^{-x^1/\beta}$. The corresponding energy is given by $\omega = \sqrt{k^2 + \lambda^2 - 1/\beta^2}$. In the subspace $k^2 + \lambda^2 < 1/\beta^2$ of the quantum numbers, the energy is imaginary and the vacuum state is unstable. Note that for a Robin plate in the Minkowski bulk the energy for the corresponding bound states is positive in the range $\beta > 1/m$. To have a stable vacuum in the AdS bulk, we assume that $\beta \leq 0$.

With the mode functions (91), the VEV of the energy–momentum tensor is evaluated by using the mode-sum formula (40) (for the procedure based on the point-splitting regularization technique see [69]). The mode-sum contains the integration over the region $k_1 \in [0, \infty)$. In the integrand, we write the parts containing the products of the trigonometric functions with the arguments $k_1 x^1 + \alpha_0(k_1)$ in the form of the sum of three terms. The first one does not depend on $x^1$ and its contribution corresponds to the VEV in

the AdS spacetime when the brane is absent. The second and third terms depend on the $x^1$-coordinate in the form of the exponents $e^{2ik_1x^1}$ and $e^{-2ik_1x^1}$. By taking into account that those terms exponentially decay in the upper and lower half-planes of the complex variable $k_1$, we rotate the integration contour over $k_1 \in [0, \infty)$ by the angle $\pi/2$ for the part with the exponent $e^{2ik_1x^1}$ and by the angle $-\pi/2$ for the part with $e^{-2ik_1x^1}$. The VEV of the energy–momentum tensor is presented in the decomposed form (41), where the diagonal components of the brane-induced part are given by the expression (no summation over $\mu$)

$$\langle T_\mu^\mu \rangle_b^{(s)} = -\frac{(4\pi)^{(1-D)/2}}{2^{2\nu+1}a^{D+1}} \int_0^\infty dx\, x e^{-2xx^1/z} \frac{\beta x/z + 1}{\beta x/z - 1}$$
$$\times \left[ A_\mu x^{D+2\nu} F_\nu^{D/2+1}(x) + \hat{B}_\mu(x) x^{D+2\nu} F_\nu^{D/2}(x) \right], \tag{93}$$

with $A_1 = 0$, $A_D = (1 - D)/2$, $A_l = 1/2$ for $l = 0, 2, \ldots, D - 1$. The operators $\hat{B}_\mu(x)$ are defined as

$$\hat{B}_l(x) = \frac{\tilde{\zeta}_1}{4} \hat{B}(x) + \frac{\tilde{\zeta}}{x} \left( \partial_x - \frac{D}{x} \right) - \tilde{\zeta}_1 \delta_{1l}, \quad l = 0, 1, 2, \ldots, D - 1,$$
$$\hat{B}_D(x) = \frac{1}{4} \hat{B}(x) - D \frac{\tilde{\zeta}}{x} \left( \partial_x - \frac{D}{x} \right) - \frac{m^2 a^2}{x^2} + \tilde{\zeta}_1, \tag{94}$$

with $\hat{B}(x) = \partial_x^2 - ((D-1)/x)\partial_x + 4$ and $\tilde{\zeta}_1$ is given by (56). In (93), we introduce the function

$$F_\nu^\mu(x) = \frac{2^{2\nu+1} x^{-2\nu}}{\Gamma(\mu - 1/2)} \int_0^1 du\, u(1 - u^2)^{\mu - 3/2} J_\nu^2(xu)$$
$$= \frac{{}_1F_2\left(\nu + 1/2; \nu + \mu + 1/2, 2\nu + 1; -x^2\right)}{\Gamma(\nu + \mu + 1/2)\Gamma(\nu + 1)}, \tag{95}$$

where ${}_1F_2(a; b, c; y)$ is the hypergeometric function. The diagonal components in the region $x^1 < 0$ are given by the expression (93) with $x^1$ replaced by $|x^1|$.

An important difference from the geometry with a brane parallel to the AdS boundary is the presence of the off-diagonal component of the vacuum energy–momentum tensor:

$$\langle T_D^1 \rangle_b^{(s)} = -\frac{(4\pi)^{(1-D)/2}}{2^{2\nu+2}a^{D+1}} \int_0^\infty dx\, e^{-2xx^1/z} \frac{\beta x/z + 1}{\beta x/z - 1} \left( \tilde{\zeta}_1 x \partial_x + 4\tilde{\zeta} \right) x^{D+2\nu} F_\nu^{D/2}(x). \tag{96}$$

This expression for the off-diagonal component in the region $x^1 < 0$ is obtained from (96) changing the sign and replacing $x^1 \to |x^1|$. Note that $\langle T_D^1 \rangle_b^{(s)} = \langle T_1^D \rangle_b^{(s)}$. Due to the nonzero off-diagonal component $\langle T_D^1 \rangle_b^{(s)}$, the Casimir force acting on the brane has two components. The first one is determined by the stress $\langle T_1^1 \rangle_b^{(s)}$ and corresponds to the component normal to the brane. The second one is obtained from $\langle T_D^1 \rangle_b^{(s)}$ and is directed along the $z$-axis. It corresponds to the shear force. Of course, because of the surface divergences in the local VEVs, both these components require an additional renormalization. Note that the Casimir forces acting tangential to the boundaries (lateral Casimir forces) may arise also in condensed matter systems if the properties of the corresponding surfaces are anisotropic or inhomogeneous (see, for example, [104,105] and references therein). In the problem under consideration, the tangential force is a consequence of the $z$-dependence of the background geometry.

From the expressions (93) and (96), it follows that for the Dirichlet and Neumann boundary conditions the brane-induced contributions to the VEV of the energy–momentum tensor differ by the signs. In these special cases the corresponding expressions are further simplified as (no summation over $\mu$)

$$
\begin{aligned}
\langle T_\mu^\mu \rangle_b^{(s)} &= \pm \frac{a^{-D-1}}{2^{D/2+\nu+1} \pi^{D/2}} [\hat{C}_\mu(u) - D\xi] f_\nu(u), \\
\langle T_D^1 \rangle_b^{(s)} &= \pm \frac{2a^{-D-1} x^1/z}{2^{D/2+\nu+1} \pi^{D/2}} \left[ \xi_1 (u-1) \partial_u^2 + (2\xi - 1) \partial_u \right] f_\nu(u),
\end{aligned}
\tag{97}
$$

where the upper and lower signs correspond to the Dirichlet and Neumann boundary conditions, respectively, and $u = 1 + 2(x^1/z)^2$. The operators $\hat{C}_\mu(u)$ in the expressions for the diagonal components are given by

$$
\hat{C}_l(u) = \xi_1 \left( u^2 - 1 \right) \partial_u^2 + \left[ 4\xi - 2 + \left( \frac{D+1}{2} \xi_1 - \frac{1}{2} \right) (u-1) \right] \partial_u,
\tag{98}
$$

for $l = 0, 2, \ldots, D-1$, and

$$
\begin{aligned}
\hat{C}_1(u) &= \xi_1 (u-1)^2 \partial_u^2 + \left( \frac{D+1}{2} \xi_1 - \frac{1}{2} \right) (u-1) \partial_u, \\
\hat{C}_D(u) &= 2\xi_1 (u-1) \partial_u^2 + \left[ 4\xi - 2 + \frac{D}{2} \xi_1 (u-1) \right] \partial_u.
\end{aligned}
\tag{99}
$$

The function $f_\nu(u)$ is expressed in terms of the hypergeometric function $_2F_1(a, b; c; x)$:

$$
f_\nu(u) = \frac{\Gamma(\nu + D/2)}{\Gamma(\nu + 1) u^{\nu + D/2}} \,_2F_1 \left( \frac{D + 2\nu + 2}{4}, \frac{D + 2\nu}{4}; \nu + 1; \frac{1}{u^2} \right).
\tag{100}
$$

For a conformally coupled massless field, one has $\nu = 1/2$ and the problem under consideration is conformally related to the problem in the Minkowski spacetime with the line element $ds^2 = \eta_{\mu\rho} dx^\mu dx^\rho$, $x^D = z$, and with planar codimension one boundaries located at $z = 0$ (the conformal image of the AdS boundary) and $x^1 = 0$ (the conformal image of the brane). The Minkowskian field obeys the Dirichlet boundary condition at $z = 0$ and the Robin boundary condition (90) at $x^1 = 0$. The Dirichlet boundary condition at $z = 0$ is a consequence of the condition we have imposed on the AdS boundary. Taking $\nu = 1/2$, from the results given above, we can obtain the VEV of the energy–momentum tensor for a conformally coupled massless field in the geometry of perpendicular planar boundaries in the Minkowski bulk by using the relation $\left\langle T_\mu^\rho \right\rangle_b^{(M)} = (a/z)^{D+1} \left\langle T_\mu^\rho \right\rangle_b$ (see (45)). In the Minkowskian limit we get the result 57 (with the replacement $y - y_0 \to x^1$) for the components $\mu = 0, 2, \ldots, D$.

Near the brane and for non-Dirichlet boundary conditions, $x^1 \ll z, |\beta|$, the leading term in the asymptotic expansions for diagonal components with $\mu = 0, 2, \ldots, D$ is given by (no summation over $\mu$)

$$
\langle T_\mu^\mu \rangle_b^{(s)} \approx - \frac{D\Gamma((D+1)/2) (\xi - \xi_D)}{2^D \pi^{(D+1)/2} (ax^1/z)^{D+1}}.
\tag{101}
$$

This coincides with the Minkowskian result where the distance from the boundary is replaced by the ratio $ax^1/z$ (compare with (58)). Note that, in accordance with (2), the latter is the proper distance from the brane measured by an observer at rest with respect to the brane. Note that the proper distance $ax^1/z$ is different from the geodesic distance $\sigma(x, x')$. The latter between the spacetime points $x = (t, 0, \mathbf{x}, z)$

and $x' = (t, x^1, \mathbf{x}, z)$ is given as $\cosh(\sigma(x, x')/a) = 1 + (x^1/z)^2/2$. For the normal stress and for the off-diagonal component near the brane one gets

$$\langle T_1^1 \rangle_b^{(s)} \approx -\frac{(x^1/z)^2}{D-1} \langle T_0^0 \rangle_b^{(s)}, \quad \langle T_D^1 \rangle_b^{(s)} \approx \frac{x^1}{z} \langle T_0^0 \rangle_b^{(s)}. \quad (102)$$

The leading terms for the Dirichlet boundary condition differ from the ones given above by the sign. The expressions (101) and (102) also describe the asymptotic behavior of the brane-induced VEV near the AdS horizon (large values of $z$ for fixed $x^1$).

Now, let us consider the asymptotics at large distances from the brane, $x^1 \gg z$. For non-Neumann boundary conditions, additionally assuming that $x^1 \gg |\beta|$, for the components $\mu = 0, \ldots, D-1$, the leading term is given by (no summation over $\mu$)

$$\langle T_\mu^\mu \rangle_b^{(s)} \approx \frac{(\xi - 1/4)(D + 2\nu) + \xi}{\pi^{D/2} \Gamma(\nu) a^{D+1} (2x^1/z)^{D+2\nu}} \Gamma(D/2 + \nu). \quad (103)$$

The asymptotics for the remaining components are expressed as

$$\langle T_D^D \rangle_b^{(s)} \approx -\frac{D}{2\nu} \langle T_0^0 \rangle_b^{(s)}, \quad \langle T_D^1 \rangle_b^{(s)} \approx \frac{D/2 + \nu}{\nu x^1/z} \langle T_0^0 \rangle_b^{(s)}. \quad (104)$$

As seen, the leading order terms for $0 < |\beta| < \infty$ coincide with those for the Dirichlet boundary condition. For a scalar field with the Neumann boundary condition ($\beta = 0$), the leading terms differ from those for the Dirichlet condition by the signs. Hence, the Dirichlet boundary condition is the attractor in a class of Robin boundary conditions with $\beta \neq 0$. At large distances, the brane-induced contributions, considered as functions of the proper distance from the brane, exhibit a power-law fall-off for both massless and massive fields. This behavior is in clear contrast with the case of the Minkowski bulk, where the boundary-induced VEVs decay exponentially for massive fields, as $e^{-2mx^1}$ (see (57)). Note that for large $x^1/z$ one has the relation $(x^1/z)^2 = \exp[\sigma(x, x')/a]$, with $\sigma(x, x')$ being the geodesic distance. For fixed $x^1$, the asymptotic formulas (103) and (104) describe the behavior of the VEVs near the AdS boundary. The diagonal components decay as $z^{D+2\nu}$, whereas the off-diagonal component tends to zero as $z^{D+2\nu+1}$. The qualitative behavior of the brane-induced energy density, as a function of the distance from the brane, is similar to what we describe in the previous section for a scalar field with the Robin boundary condition on the brane parallel to the AdS boundary.

## 6. Summary

We consider the influence of a brane in AdS bulk on the properties of quantum vacuum. Two geometries are discussed: (i) a brane parallel to the AdS boundary; and (ii) a brane perpendicular to the AdS boundary. In the first geometry, as a local characteristic of the vacuum state, the VEV of the energy–momentum tensor is investigated for scalar, Dirac and electromagnetic fields. For calar field, a general Robin boundary condition is considered and the Dirac field is constrained by the bag boundary condition. In the case of the electromagnetic field, two types of boundary conditions are discussed. The first one corresponds to the perfect conductor boundary conditions in 3D electrodynamics and the second one is the analog of the boundary condition used in bag models of hadrons to confine the gluons. The VEV of the energy–momentum tensor is expressed as a mode-sum over complete set of mode functions and for all these cases the corresponding sets are given. The brane divides the background geometry into two regions: the region between the brane and AdS horizon (R-region) and the region between the brane and AdS boundary (L-region). Although the AdS spacetime is homogeneous, the brane has

a nonzero extrinsic curvature tensor and the properties of the quantum vacuum in those regions are different. In particular, the spectrum of the quantum number $\lambda$, corresponding to the momentum along the direction normal to the AdS boundary, is continuous in the R-region and discrete in the L-region. In the latter region, the eigenvalues are zeros of cylinder functions. The mode-sum for the VEV of the energy–momentum tensor contains series over those zeros and for the summation we have employed the generalized Abel–Plana formula. That allows extracting from the VEV the part corresponding to the geometry without a brane and to present the brane-induced contribution in terms of integral, exponentially convergent for points away form the brane. A similar decomposition is provided for the R-region.

Near the brane, the leading terms in the asymptotic expansions for the energy density and parallel stresses coincide with the corresponding expressions for a single boundary in the Minkowski bulk, where the distance from the boundary is replaced by the proper distance from the brane on the AdS bulk. For those VEVs, the effect of gravity is weak. This is related to the fact that, near the brane, the main contribution to the corresponding VEVs come from the vacuum fluctuations with the wavelengths smaller than the curvature radius of the background geometry and influence of the gravitational field on those modes is weak. For a boundary in the Minkowski bulk, the normal stress is zero. The nonzero normal stress in the geometry of a brane on the AdS bulk is a purely gravitational effect. The effect of gravity on the brane-induced VEVs is essential at distances from the brane larger than the curvature radius. In particular, for the R-region, at large distances, the decay of the brane-induced contribution in the vacuum energy–momentum tensor, as a function of the proper distance, is exponential for both massless and massive fields. For the Minkowski bulk and for massless fields, the fall-off of the boundary-induced contribution is as power-law. On the AdS boundary, the brane-induced contributions tend to zero as $z^{D+\beta}$, where $\beta = 2\nu$ for scalar field (with $\nu$ given by (7)), $\beta = 2ma + 1$ for the Dirac field and $\beta = D - 2$ for the electromagnetic field. Near the AdS boundary, one has a simple relation between the energy density and the normal stress, given by $\langle T_D^D \rangle_{\rm b} \approx -(D/\beta)\langle T_0^0 \rangle_{\rm b}$. This correspond to the barotropic equation of state for the vacuum pressure $-\langle T_D^D \rangle_{\rm b}$ along the $z$-direction and vacuum energy density. Note that, for the pressures along the directions parallel to the brane, the equation of state is of the cosmological constant type. By using the generalized zeta function technique, we also investigate the VEV of the surface energy–momentum tensor. From the viewpoint of the observer living on the brane, the latter corresponds to a gravitational source of cosmological constant type. Depending on the value of the coefficient in the boundary condition, the induced cosmological constant can be either positive and negative.

The brane-induced effects on the quantum vacuum for Geometry (ii), we consider the example of a scalar field with general curvature coupling parameter. For the Robin boundary condition the mode functions have the form (91). The diagonal components of the brane-induced energy–momentum tensor are given by the expressions (93). An important difference from the problem with a brane parallel to the AdS boundary is the presence of nonzero off-diagonal component (96) of the vacuum energy–momentum tensor. As a consequence, the Casimir force acting on the brane, in addition to the normal component, contains a component directed parallel to the brane (shear force). At large distances from the brane, the decay of the brane-induced contribution to the energy–momentum tensor, as a function of the proper distance from the brane, is as power-law for both massive and massless field. As mentioned above, in the Minkowski bulk, the decay for massive fields is exponential.

For charged fields, another important local characteristic of the vacuum state is the expectation value of the current density. This VEV in models with local AdS geometry and with toroidally compact spatial dimensions, in the presence of single and two branes has been investigated in [72,76,106,107] for charged scalar and fermionic fields. The vacuum currents have nonzero components along the compact dimensions only. They are periodic functions of the magnetic flux with the period equal to the flux quantum. Depending on the boundary conditions imposed on the fields at the locations of the branes,

the brane-induced effects lead to increase or decrease of the current density. Applications are discussed to Randall–Sundrum type braneworld models as well as curved graphene tubes.

In the discussion above, we assume that the background geometry is fixed. Among the interesting directions for the further research is the investigation of the back-reaction of quantum effects on the geometry by using the semiclassical Einstein equations with the VEV of the energy–momentum tensor in the right-hand side. The vacuum energy–momentum tensor may violate the energy conditions in the singularity theorems, and this leads to interesting cosmological dynamics of the bulk and on the brane. In this regard, the next step in the study of local quantum effects in braneworlds could be the investigation of the vacuum energy–momentum tensor in models with dS branes.

**Funding:** This research received no external funding.

**Acknowledgments:** The author is grateful to Stefano Bellucci, Eugenio Bezerra de Mello, Emilio Elizalde, Sergei Odintsov, Mohammad Setare and Valery Vardanyan for collaboration.

**Conflicts of Interest:** The author declares no conflict of interest.

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
