# Peer review of "Quantum Vacuum Effects in Braneworlds on AdS Bulk"

_universe, doi:10.3390/universe6100181_

Round 1

Reviewer 1 Report

This paper reviews the results for the vacuum expectation values of the energy-momentum tensor for quantum fields of different spin in anti-de Sitter (AdS) spacetime in the presence of branes. In the first part, the vacuum energy-momentum tensor is considered for a brane parallel to the boundary of the AdS spacetime. For scalar, Dirac and electromagnetic fields with appropriate boundary conditions, the brane-induced contributions are separated and their behaviors in asymptotic regions of the parameters are studied. It is shown that in all these cases the influence of the gravitational field on the boundary-induced contributions is essential at distances from the brane larger than the curvature radius of the background spacetime. In the second part of the paper the brane is perpendicular to the AdS boundary and its influence on the vacuum energy-momentum tensor is discussed for a scalar field with general curvature coupling parameter. The corresponding geometryis less symmetric than the one discussed in the first part and the energy-momentum tensor for non-conformally coupled fields is not diagonal. As a consequence, the Casimir force acting on the brane, in addition to the normal component, has a nonzero component parallel to the brane which is in some analogy to the lateral Casimir force acting between corrugated surfaces [Chiu et al., PRB 80, 121402(R) (2009); 81, 115417 (2010)].

The paper can be recommended for publication after highlighting this analogy and taking into account the following comments.

1. All the discussed results are related to the expectation value for the bulk energy-momentum tensor. It is mentioned, however, that, in addition to the bulk tensor, there is also surface energy-momentum tensor that is located on the brane. Several references are given where the corresponding expectation value is investigated. For a completeness of this review, it would be desirable to add a more detailed presentation of this point.

2. Another maximally symmetric curved background is the de Sitter spacetime, and I recommend to add a brief summary of the corresponding results for a planar boundary in that geometry.

Reviewer 2 Report

I think this paper is generally acceptable for publication.  I'm not sure that it should be classified as a review, however, since it seems to be mostly reporting on original research.  But the work of certainly of theoretical interest.

I do have some corrections and queries:

l. 46, the word should be "fields"

The gauge condition referred to 2 lines below (2.28) is the "Lorenz" gauge not the "Lorentz" gauge.

l. 188: The fact that the regularization is completely implicit is a bit worrisome.  In particular, without regularization, ambiguities can arise.  The author should address this.

4 lines below (3.3): "base" should be "basis"

The sentence including (3.5): The author should address the trace and divergence anomalies that generally occur in curved space.

l. 209: The T^\mu_\mu notation is not good, since it implies a trace.  Maybe T^{\mu\mu}  would be better (with some metric adjustments).

2 lines before (3.10): The rotation to imaginary frequencies in two different senses seems dubious.  This needs to be explained.

The labelling in Fig. 1 is confusing.  At least the upper label might be \mu=D.  (D is used in a completely difference sense in Fig. 2.)

l. 248 and following. In the remarks about Fig. 2a, the sign change and nonmonotonicity should be discussed.

The different curves in Fig. 2 need to be individually labelled.  The three intermediate curves in Fig. 2a should be denoted by the values of \beta/a, and which curves belonging to which label need to be distinguished in Fig. 2b.  Note D=Dirichlet here, not the Dth coordinate.

The same remarks on labelling need to be applied to Fig. 3.

2 lines before (3.23): the second equation cited should be (2.32).

Fig. 4, again there is a labelling problem, because D=Dirichlet was used earlier.  Maybe it should be \mu=0, \mu=D in each case.

line after l. 296: "ration" should be "ratio"

Fig. 5: again \mu=D would be less confusing.  Fig. 5b: it looks almost as if the curves tend to zero as ma to 0, but I think not.  This might be stated more explicitly.

Fig. 6: again the labelled of the different curves needs to be fixed.

It looks like there is a sign discrepancy between (3.38) and (3.39), since it is stated one force is attractive, the other repulsive.  (Boyer effect.)

Fig. 7: Once more a labelling problem: \mu=0, \mu=D.

l. 356: Again, a rotation through two difference senses.  This doesn't sound right.

l. 453: the word "be" should be inserted before "present"

Overall, I found this an interesting read.  It is of technical interest, although it is not clear that there are significant physical implications of the results.

Reviewer 3 Report

In this paper, the author reviews results on the vacuum expectation value of the energy-momentum tensor in the presence of a D-brane in AdS space for scalar, electromagnetic and Dirac fields. The cases of the D-brane parallel and orthogonal to the boundary of the AdS are analysed in detail and the Robin and bag boundary conditions are thoroughly discussed. 

I believe that the paper is an interesting contribution to the literature and I recommend it for publication.

Reviewer 4 Report

The author considers in this paper the influence of a brane in an AdS bulk on the properties of the corresponding quantum vacuum. Two geometries are discussed, namely a brane parallel to the AdS boundary and a brane perpendicular to the AdS boundary.

The paper is well written, systematically and in good detail. It is informative and useful for readers working on these subjects. I did not find any errors or mistakes.

The English is good enough but could be improved in some places.

Concerning the references I just missed a couple of papers that are rather well known and very much in connection with the research carried out here:

E. Elizalde, S. Nojiri, S.D. Odintsov and S. Ogushi,
Casimir effect in de Sitter and anti-de Sitter braneworlds,
Phys. Rev. D67 (2003) 063515 [arXiv:hep-th/0209242].

Reviewer 5 Report

In this paper, the author has studied "Quantum vacuum effects in braneworlds on AdS bulk". My specific criticism regarding this paper is appended below:

  1. The presentation and the overall physical contents of the paper are perfectly presented.
  2. Computations are correct and contribute significantly to the literature.
  3. Only one point I want to point out that the reference list of the present version of the paper is not at all-sufficient. There are many more works that exist in this literature. I can point a few of them, . arXiv:hep-ph/9907447,  arXiv:hep-ph/9911457,  arXiv:hep-th/0310007 , arXiv:1405.6826 [hep-th], hep-th/0603014 [hep-th], arXiv:1209.3978 [hep-th], JHEP08(2003)053, JHEP 1302 (2013) 136 and there are many more. If the author can include many more such references including the pointed references it would give the readers a complete understanding of the associated literature.
  4. The conclusion of this paper can be presented with a few more interesting future prospects in a better way. So in this connection, I suggest the author to incorporate some minor changes in the revised version.

Once the above mentioned minor corrections are implemented properly, then I will give my final decision regarding the publication of this paper in Universe. 
